# SPARK: Synergistic Policy And Reward co-evolving framework

## Abstract

Recent Large Language Models (LLMs) and Large Vision-Language Models (LVLMs) increasingly use Reinforcement Learning (RL) for post-pretraining, such as RL with Verifiable Rewards (RLVR) for objective tasks and RL from Human Feedback (RLHF) for subjective tasks. However, RLHF incurs high costs and potential reward–policy mismatch due to reliance on human preferences, while RLVR still wastes supervision by discarding rollouts and correctness signals after each update. To address these challenges, we introduce the **S**ynergistic **P**olicy **A**nd **R**eward Co-Evolving Framewor**K** (SPARK), an efficient, on-policy, and stable method that builds on RLVR. Instead of discarding rollouts and correctness data, SPARK recycles this valuable information to simultaneously train the model itself as a generative reward model. This auxiliary training uses a mix of objectives, such as pointwise reward score, pairwise comparison, and evaluation conditioned on further-reflection responses, to teach the model to evaluate and improve its own responses. Our process eliminates the need for a separate reward model and costly human preference data. SPARK creates a positive co-evolving feedback loop: improved reward accuracy yields better policy gradients, which in turn produce higher-quality rollouts that further refine the reward model. Our unified framework supports test-time scaling via self-reflection without external reward models and their associated costs. We show that SPARK achieves significant performance gains on multiple LLM and LVLM models and multiple reasoning, reward models, and general benchmarks. For example, SPARK-VL-7B achieves an average 9.7% gain on 7 reasoning benchmarks, 12.1% on 2 reward benchmarks, and 1.5% on 8 general benchmarks over the baselines, demonstrating robustness and broad generalization.

## 1 Introduction

Reinforcement learning (RL) is a standard step of post-pretraining improvement and alignment for Large Language Models (LLMs) and Large Vision-Language Models (LVLMs). In practice, current RL systems rely on two complementary routes: **(1)** RL with verifiable rewards (**RLVR**) (Lambert et al., 2024a; Guo et al., 2025; Team et al., 2025), which uses a verifier to address objective and verifiable problems like math and code. **(2)** Reward-model–based pipelines such as RL from Human Feedback (**RLHF**) (Ouyang et al., 2022; Bai et al., 2022), which distill human or synthetic preferences into a learned reward model to guide policy optimization on subjective tasks. These two RL stages have yielded significant gains in reasoning quality, safety, and truthfulness, and have become a cornerstone in modern LLM/LVLM training.

Despite impressive progress, current RL pipelines for LLMs/LVLMs still exhibit several limitations. Approaches based on verifiable rewards (RLVR) are effective only for tasks with explicit verifiers, leaving open-ended objectives like helpfulness and safety unaddressed. Conversely, reward-model–based pipelines (RLHF) can handle subjective tasks with reward models (Su et al., 2025a) or LLM-as-a-judge (Zheng et al., 2023; Gunjal et al., 2025) but demand substantial and costly curated human preference data. Furthermore, training the reward model as a separate component causes it to lag the evolving policy, inducing reward-policy mismatch, reward hacking, and brittle generalization under out-of-distribution queries (Skalse et al., 2022; Gao et al., 2023). Finally, dependence on external reward models or judge models introduces significant latency and serving costs during both training and test-time scaling (Zhao et al., 2025).

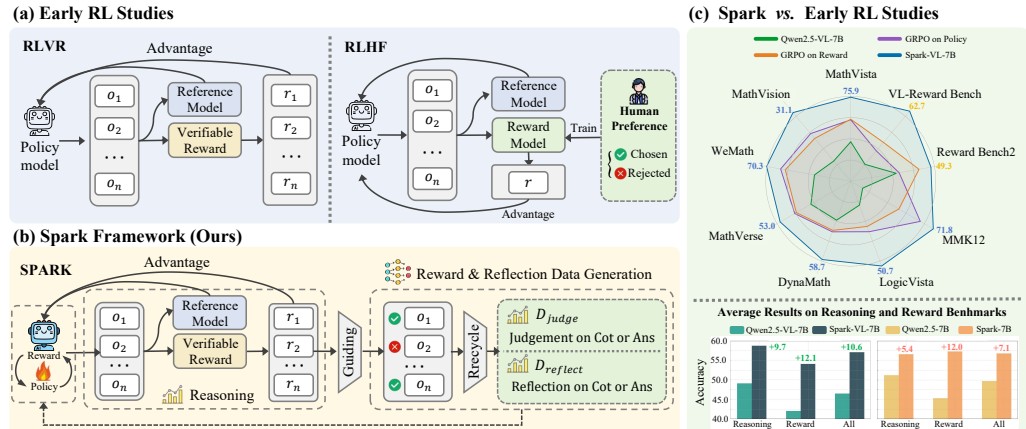

Figure 1: **(a)** Early studies of RL with Verifiable Rewards (RLVR) and RL from Human Feedback (RLHF) that rely on reward models. **(b)** We propose SPARK that recycles the rollouts from the RLVR, and further trains the model itself as a generative reward model. **(c)** SPARK consistently outperforms early RL approaches in both reasoning and reward model benchmarks.

To mitigate the limitations of early RL studies, such as costs of human preference labeling and deployment, we turn to an *internalized* source of supervision. Our method builds on RL with Verifiable Rewards (RLVR), where $n$ candidate responses or rollouts $\{o_1, o_2, \ldots, o_n\}$ are generated, and score them against a ground-truth label to update the policy model (see Fig. 1 **(a)**). However, these valuable rollouts are typically **discarded** after this single use. Our key insight is to **recycle** the rollouts and correctness data to further train the model itself as a generative reward model simultaneously. We use the RLVR-derived correctness scores to train the model on a mix of objectives: a pointwise objective to determine if a response is correct, a pairwise objective to identify which response is better, and a reflection objective to learn how to fix an incorrect response to get the correct one.

The proposed *auxiliary* training paradigm for RLVR, the **S**ynergistic **P**olicy **A**nd **R**eward Co-Evolving Framewor**K** (**SPARK**), enhances reward accuracy, yielding stronger policy gradients and improving the model's reasoning abilities (see Fig. 1 **(b)**). We further use this internal judge for self-reflection at test time, extending alignment to tasks beyond strictly verifiable domains while retaining the robustness of verifiable feedback. SPARK has four advantages: (1) **Data- and compute-efficient**: no extra human preference data annotation or separate reward model training loop is required, as the signals come "for free" from RLVR training rollouts. (2) **On-policy and stable**: reward data are continually sampled from and calibrated to the model's current behavior, reducing reward–policy mismatch. (3) **Co-evolving**: improved reward accuracy yields better gradients for the policy, which produces high-quality rollouts, further refining the reward. (4) **Unified development**: our framework enables RL training and test-time scaling, removing the dependency on an external reward model, and thereby saving GPU memory and reducing the communication overhead.

Our SPARK is applicable to both LLMs (e.g., Qwen2.5 (Yang et al., 2025a)) and LVLMs (e.g., Qwen2.5-VL (Bai et al., 2025)). As shown on Fig. 1 **(c)**, SPARK achieves clear improvements on various mathematical reasoning and reward model benchmarks. For LVLMs, SPARK-VL-7B improves by **9.7%** on 7 reasoning and **12.1%** on 2 reward benchmarks, in addition to an average **1.5%** gain on 8 general benchmarks. These improvements are observed also with larger LVLM models (SPARK-VL-32B) and pure LLMs (SPARK-7B), demonstrating the robustness across different model scales and architectures.

Our key contributions are: **(1)** We introduce an efficient, on-policy, and stable framework SPARK that builds on RLVR but recycles the valuable rollouts that are typically discarded after policy updates. We use the RLVR-derived correctness scores to train the model itself to become a generative reward model, which eliminates the need for human preference data to train a separate, external reward model with additional development costs. **(2)** Our SPARK is designed as a **co-evolving** mechanism. Improved reward accuracy yields better gradients for the policy, which in turn produces higher-quality rollouts. These high-quality rollouts further refine the reward model, creating a positive feedback loop that leads to stronger overall performance and stability. **(3)** Extensive experiments show that SPARK achieves substantial improvements on multiple LVLM and LLM models. SPARK-

VL-7B achieves average **9.7%** gains on 7 reasoning benchmarks, **12.1%** on 2 reward benchmarks, and **1.5%** on 8 general benchmarks, demonstrating the strong generalization ability.

## 2 RELATED WORKS

**Reinforcement Learning with Verifiable Reward.** Following the success of DeepSeek-R1 (Guo et al., 2025), the GRPO (Shao et al., 2024) algorithm—driven by verifiable rewards—has demonstrated strong potential across a variety of reasoning-intensive tasks, particularly in mathematics and programming. Moreover, this RLVR paradigm has been successfully extended to a wide range of domains, including perception (Zheng et al., 2025; Su et al., 2025b; Liu et al., 2025a; Peng et al., 2025), agent (Jin et al., 2025; Liu et al., 2025b) and so on. In this work, we adopt a GRPO-based algorithm to build our synergistic policy and reward co-evolving framework. Through reinforcement learning, our framework jointly enhances the policy's reasoning and the reward's judging abilities in a unified model, breaking the isolation between policy and reward models in prior approaches.

**Reinforcement Learning from Human Feedback.** Reinforcement Learning from Human Feedback (RLHF) optimizes policy models using human preference data. These data are either directly collected from human annotations or generated by teacher models, and are typically used to first train an independent reward model. The reward model then provides feedback signals that guide policy optimization (Cai et al., 2024; Zhu et al., 2023; Zang et al., 2025; Kim et al., 2023; Yuan et al., 2024; Lambert et al., 2024b; Ivison et al., 2023). However, a key limitation of existing paradigms is that policy and reward models are usually developed in isolation, which restricts their interaction and reduces the potential for mutual improvement. In this work, we instead treat policy and reward as complementary capabilities, and introduce SPARK, a unified framework where the two evolve jointly, reinforcing each other and ultimately achieving stronger overall performance. We also discuss related work on self-reward and self-reflection; please refer to Appendix. A.3.

## 3 METHODS

In this section, we provide a detailed introduction to the SPARK approach. Specifically, Sec. 3.1 presents the SPARK training framework with verifiable reward, Sec. 3.2 outlines the on-policy reward&reflection data generation of SPARK, and finally, Sec. 3.3 details the test-time scaling evaluation strategy used in SPARK.

### 3.1 SPARK TRAINING WITH VERIFIABLE REWARD

Fig. 2 (a) illustrates our Synergistic Policy and Reward Co-Evolving Framework. In contrast to prior approaches, our method integrates the training of policy and reward into a unified framework, where both components are optimized within a single model under the guidance of verifiable rewards. In this section, we detail how SPARK employs verifiable rewards to guide optimization during training, enabling the model to co-evolve its policy and reward capabilities. Through this process, the model develops not only into a strong reasoning system but also into an effective reward model.

**Step 1: Sampling an answer group.** As shown in Fig. 2(a), given a Visual Question Answering (VQA) sample $d = (q, a, I)$, or $d = (q, a)$ in the case of a language-only LLM without visual input, the model generates an answer group of size $n$, denoted as

$$G = \left\{ \left( o_{\text{cot}}^i, o_{\text{ans}}^i \right) \right\}_{i=1}^n, \tag{1}$$

where $q$ denotes the input prompt, $a$ the ground-truth answer, $I$ the input image, $o_{\text{cot}}^i$ the $i$-th reasoning trace, and $o_{\text{ans}}^i$ the corresponding final answer. To facilitate a clear separation between reasoning and the final output, we design a prompt that requires the answer to be enclosed in \box{}, as illustrated in Appendix A.2.

**Step 2: Verifiable reward.** Each final answer is evaluated by a rule-based, verifiable reward:

$$\mathbb{R}(q, o) = \begin{cases} 1, & \text{if } o = a, \\ 0, & \text{otherwise.} \end{cases} \tag{2}$$

For the $i$-th sample in the answer group $G$, we denote its reward as $r^i = \mathbb{R}\left(q, o_{\text{ans}}^i\right)$.

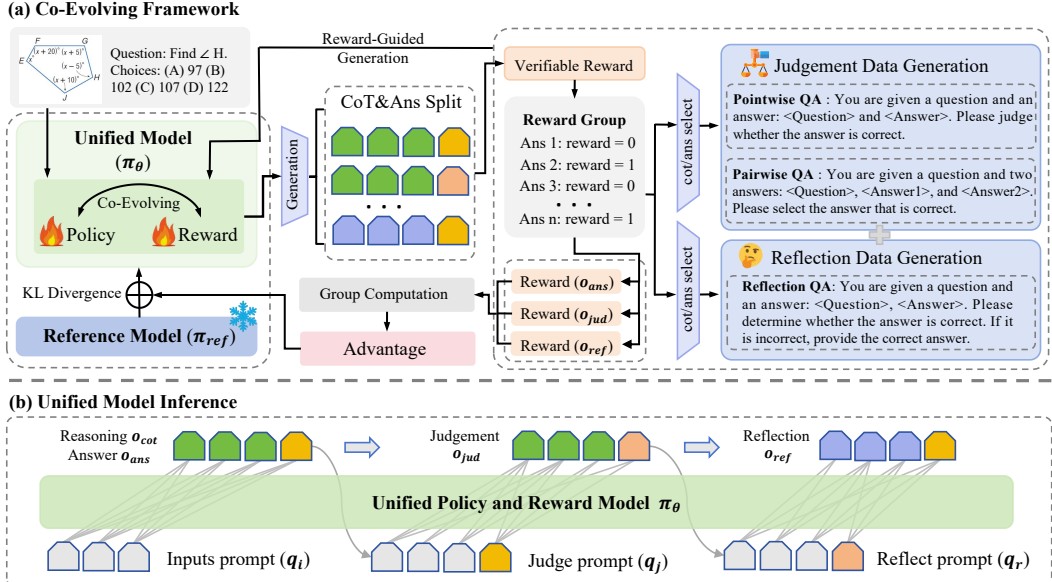

Figure 2: **SPARK Framework.** (a) **Training**: Our method recycles the valuable rollouts from verifiable reward-guided generation to simultaneously train a unified policy model $\pi_\theta$, also as a generative reward model. (b) **Inference**: at test time, the single unified model can handle reasoning, judgment, and reflection tasks for test-time scaling, eliminating the need for external reward or judge models.

**Step 3: Advantage computation.** Following GRPO-style normalization, we compute a standardized advantage for each candidate:

$$\bar{r} = \frac{1}{n}\sum_{j=1}^{n} r^j, \qquad s = \sqrt{\frac{1}{n}\sum_{j=1}^{n}\big(r^j - \bar{r}\big)^2 + \epsilon}, \qquad A^i = \frac{r^i - \bar{r}}{s}, \tag{3}$$

where $\bar{r}$ is the mean reward across the $n$ candidates, $s$ is the standard deviation with a small constant $\epsilon > 0$ for numerical stability, and $A^i$ is the normalized advantage used for policy gradient updates.

**Step 4: Overall objective.** The training objective maximizes the expected verifiable reward while regularizing the learned policy $\pi_\theta$ towards a reference policy $\pi_{\text{ref}}$:

$$\mathbb{E}_{o\sim\pi_\theta(\cdot\,|\,q)}\big[\mathbb{R}(q, o)\big] - \lambda\,\mathrm{KL}\big(\pi_\theta(\cdot\,|\,q)\,\|\,\pi_{\text{ref}}(\cdot\,|\,q)\big), \tag{4}$$

where $\lambda$ is a hyperparameter controlling the KL-divergence. This formulation ensures that the optimization signal comes directly from task-defined correctness, enabling efficient and stable training whenever outputs are objectively verifiable.

## 3.2 SPARK ON-POLICY DATA GENERATION

Unlike early RL methods that optimize only the policy model, the advantage of SPARK lies in the joint co-evolution of policy and reward within a single model. To achieve this, SPARK generates reward and reflection data on-policy during the reasoning process, which requires neither additional preference annotations nor teacher models, making it highly efficient.

Beyond computing the advantage for gradient optimization in Eq. 3, the reward values $r^i$ also guide the on-policy generation of reward and reflection data, as illustrated in Fig. 2. Specifically, we categorize the generated data into three forms: pointwise, pairwise, and reflection, each contributing to different aspects of judgment and self-reflection.

**Pointwise.** We reorganize $(q, G)$ into binary judgment samples of the form

$$\mathbb{D}_{\text{pointwise}} = \big\{\big(q,\ \boldsymbol{o}^i_{\text{ans}},\ \mathbb{R}(q, \boldsymbol{o}^i_{\text{ans}})\big)\big\}, \tag{5}$$

where the model is asked to determine whether a single candidate answer $o^i_{\text{ans}}$ is correct. This formulation directly trains the model's ability to judge the validity of individual answers.

**Pairwise.** We also construct comparison-style samples:

$$\mathbb{D}_{\text{pairwise}} = \left\{ \left( q, \; \boldsymbol{o}_{\text{ans}}^i, \; \boldsymbol{o}_{\text{ans}}^j, \; \mathbb{R}(q, \boldsymbol{o}_{\text{ans}}^i), \; \mathbb{R}(q, \boldsymbol{o}_{\text{ans}}^j) \right) \right\}, \tag{6}$$

where two candidate answers $o_{\text{ans}}^i$ and $o_{\text{ans}}^j$ are drawn from $G$, and the model must select the better one. This encourages preference-style judgment, allowing the model to distinguish between relatively stronger and weaker outputs. Notably, in both pointwise and pairwise settings, $o_{\text{ans}}$ can be replaced with reasoning traces $o_{\text{cot}}$ to shift supervision toward intermediate steps.

**Reflection.** Finally, we construct reflection-style samples:

$$\mathbb{D}_{\text{reflect}} = \left\{ \left( q, \; \boldsymbol{o}_{\text{ans}}^i, \; \mathbb{R}(q, \boldsymbol{o}_{\text{ans}}^i) \right) \right\}, \tag{7}$$

where the model first verifies correctness and, if $R(q, o_{\text{ans}}^i) = 0$, the incorrect answer is then fed back to the model for reflection and refinement. This process explicitly stimulates the model's self-reflection capability.

The combined dataset is then given by

$$\mathbb{D}_{\text{on-policy}} = \mathbb{D}_{\text{pointwise}} \; \cup \; \mathbb{D}_{\text{pairwise}} \; \cup \; \mathbb{D}_{\text{reflect}}, \tag{8}$$

which is used to further optimize the unified policy–reward model, strengthening both its judgment and self-reflection abilities. Representative prompt templates used for data generation are provided in Appendix. A.2.

### 3.3 TEST TIME SCALING WITH SELF-REFLECTION

Benefiting from the co-evolution of policy and reward capabilities, SPARK functions not only as a strong policy model but also as a strong reward model. The synergy between these two abilities further enhances the model's capacity for self-reflection, which proves especially valuable in the context of test-time scaling (TTS).

As illustrated in Fig. 2(b), we adopt a TTS procedure to evaluate the model's reasoning, judgment, and reflection abilities. Formally, given a question $q$ and image $I$, the model generates a candidate answer at step $t$ as

$$\mathbf{o}_t = \pi_{\boldsymbol{\theta}}(q, I), \tag{9}$$

where $\pi_\theta$ denotes the model, and $o_t = (c_t, a_t)$ consists of a reasoning chain $c_t$ and a final prediction $a_t$. The model then assesses its own output through a judgment prompt:

$$\mathbf{r}_t = \pi_{\boldsymbol{\theta}}\big(q, I, \text{judge}(c_t, a_t)\big), \quad \mathbf{r}_t \in \{0, 1\}, \tag{10}$$

where $\text{judge}(c_t, a_t)$ instructs the model to verify whether $(c_t, a_t)$ is correct.

Based on this evaluation, the model either accepts the result or performs iterative refinement:

$$\mathbf{o}_{t+1} = \begin{cases} \mathbf{o}_t, & \text{if } \mathbf{r}_t = 1, \\ \pi_{\boldsymbol{\theta}}\big(q, I, \text{reflect}(c_t, a_t)\big), & \text{if } \mathbf{r}_t = 0, \end{cases} \tag{11}$$

where $\text{reflect}(c_t, a_t)$ prompts the model to critique its prior reasoning and generate a revised solution. The process terminates once the model produces an answer it judges correct, and accuracy is computed by comparing this final prediction with the ground truth.

## 4 EXPERIMENTS

### 4.1 EXPERIMENTAL SETUP

**Benchmarks** To comprehensively evaluate the effectiveness of SPARK, we conduct experiments on three categories of benchmarks: mathematical, reward-related, and general. For mathematical benchmarks, we assess both multimodal and language-only reasoning using representative datasets such as MathVista (Lu et al., 2023) and GSM8k (Cobbe et al., 2021). For reward-related evaluation, we employ RewardBench2 (Malik et al., 2025) and VL-RewardBench (Li et al., 2025), including their mathematical subsets for fine-grained analysis. For general capabilities, we test on widely used multimodal benchmarks such as MMBench (Liu et al., 2023) and MMStar (Chen et al., 2024). For the complete list of benchmarks used in this work, please refer to Sec. A.1.2.

Table 1: **Evaluation Results on SPARK-VL-7B.** We evaluate SPARK on multiple mathematical and reward-related benchmarks. Here, RB2 denotes RewardBench2, and VL-RB denotes VL-RewardBench.

| Model | VLM Math Benchmark | | | | | | | Avg-M | Reward Benchmark | | | | Avg-R | Avg-All |
|---|---|---|---|---|---|---|---|---|---|---|---|---|---|---|
| | MathVista | MathVision | WeMath | MathVerse | DynaMath | LogicVista | MMK12 | | RB2 | RB2-Math | VL-RB | VL-RB-Math | | |
| *Baseline* | | | | | | | | | | | | | | |
| Qwen2.5-VL-7B | 68.2 | 25.1 | 62.1 | 49.2 | 53.3 | 40.4 | 45.1 | 49.1 | 45.8 | 38.8 | 47.7 | 35.5 | 42.0 | 46.5 |
| OpenVLThinker-7B | 70.2 | 25.3 | 64.3 | 47.9 | - | - | 60.6 | - | 48.5 | 37.3 | 33.2 | 33.1 | 38.0 | - |
| Vison-R1-7B | 73.5 | 27.4 | **75.0** | 52.4 | 54.9 | 37.1 | 36.7 | 51.0 | 32.6 | 28.1 | - | - | - | - |
| R1-OneVision-7B | 64.1 | _29.9_ | 61.8 | 47.1 | - | 39.1 | 39.8 | - | 35.7 | 33.1 | 37.6 | 37.4 | 36.0 | - |
| VL-ReThinker-7B | 73.7 | 28.4 | 67.9 | **54.0** | _57.3_ | 42.7 | 64.9 | 55.6 | 42.3 | 35.0 | 47.1 | 32.5 | 39.2 | 49.6 |
| MM-Eureka-7B | 73.0 | 26.9 | 66.1 | 50.3 | 56.9 | _48.9_ | 64.5 | 55.2 | 44.9 | _41.0_ | 48.8 | 36.8 | 42.9 | 50.7 |
| *Qwen2.5-VL-7B + GRPO* | | | | | | | | | | | | | | |
| + Policy-Only | 72.0 | 28.5 | 67.9 | 51.2 | 54.9 | 44.9 | 66.9 | 55.2 | 46.1 | 40.4 | 51.5 | 62.1 | 50.0 | 53.3 |
| + Reward-Only | 72.1 | 27.9 | 67.1 | 51.0 | 54.7 | 44.0 | 58.8 | 53.7 | 48.1 | 39.3 | 53.9 | 62.7 | 51.0 | 52.7 |
| + Policy&Reward | _74.2_ | 28.9 | _70.9_ | 51.3 | 56.3 | 46.2 | 67.9 | _56.5_ | _48.9_ | **43.7** | _54.4_ | 63.9 | _52.7_ | _55.1_ |
| *Ours* | | | | | | | | | | | | | | |
| SPARK-VL-7B | **75.9** | **31.1** | 70.3 | _53.0_ | **58.7** | **50.7** | **71.8** | **58.8** | **49.3** | 39.2 | **62.7** | **65.1** | **54.1** | **57.1** |
| Δ | +7.7 | +6.0 | +8.2 | +3.8 | +5.4 | +10.3 | +26.7 | +9.7 | +3.5 | +0.4 | +15.0 | +29.6 | +12.1 | +10.6 |

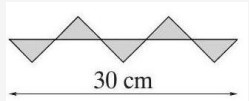

The figure in the drawing consists of five isosceles right triangles of the same size. Find the area (in $cm^2$) of the shaded figure. Please first conduct reasoning, and then answer the question. Repeat the final answer using a '\\boxed{}'.

**Reasoning Trace:**
To determine the area of the shaded figure, we start by analyzing the given information and the geometric configuration......Since the triangles are arranged in a way that their hypotenuses form the base of the figure, the total length of the base is \\( 5x \\).\n\nThus, we have:\n\\[ 5x = 30 \\]\nSolving for \\( x \\), we get:\n\\[ x = 6 \\]\n\nThe area of one isosceles right triangle is given by:\n\\[ \\text{Area} = \\frac{1}{2} \\times x \\times x = \\frac{1}{2} \\times 6 \\times 6 = 18 \\, \\text{cm}^2 \\]\n\nSince there are five such triangles, the total area of the figure is:\n\\[ 5 \\times 18 = 90 \\, \\text{cm}^2 \\]\n\nHowever, the shaded figure consists of three of these triangles. Therefore, the area of the shaded figure is:\n\\[ 3 \\times 18 = 54 \\, \\text{cm}^2 \\]\n\nThe final answer is:\n\\[ **boxed{54}** \\]

**Judgement:**
The given answer is 54 cm², which is incorrect. Thus, the judgment is \\**boxed{No}**.

**Self-Reflection:**
1. Since the figure consists of five isosceles right triangles of the same size, let's denote the length of the legs of each triangle as \\( x \\). The hypotenuse of each triangle will be \\( x\\sqrt{2} \\).\n\n......4. Since there are five such triangles, the total area of the figure is:\n  \\[ 5 \\times 18 = 90 \\text{ cm}^2\n  \\]\n\n5. However, the figure is divided into two equal parts by the central line, and the shaded part is half of the total area of the figure. Therefore, the area of the shaded figure is:\n  \\[\n  \\frac{90}{2} = 45 \\text{ cm}^2\n  \\]\n\nThe correct final answer is:\n\\[\n\\**boxed{45}**\n\\]

Figure 3: **Math Reasoning Case.** We illustrate the reasoning process of SPARK on a mathematical task, covering reasoning, judgment, and reflection. For brevity, parts of the content are omitted.

**Baseline Methods** Our experiments are built upon the Qwen2.5-VL (Bai et al., 2025) and Qwen2.5 (Yang et al., 2025a) model series. For comparison, we include representative RL-based baselines such as VL-Rethinker (Wang et al., 2025), MM-Eureka (Meng et al., 2025), Vision-R1 (Huang et al., 2025), as well as the standard GRPO baselines, where *Policy-Only* and *Reward-Only* denote models trained to improve reasoning or judgment in isolation. Full details of the baseline models are provided in Sec. A.1.1 of the supplementary material.

## 4.2 RESULTS ON MATHEMATICAL AND REWARD BENCHMARKS

**Results on SPARK-VL-7B.** Tab. 1 reports the results of SPARK-VL-7B on both mathematical and reward-related benchmarks. Compared with the Qwen2.5-VL-7B baseline, our model achieves consistent and substantial gains. Specifically, SPARK delivers an average improvement of **9.7%** on mathematical benchmarks and **12.1%** on reward benchmarks, resulting in an overall gain of **10.6%**.

The *Qwen2.5-VL-7B+GRPO* ablations further provide insight into the effect of different training signals. Training with *Policy-Only* data slightly favors mathematical reasoning, while *Reward-Only* training better enhances judgment ability. When both sources are combined (+*Policy&Reward*), the model surpasses either single-source variant, indicating the complementary of policy and reward supervision. Building on this, SPARK-VL-7B advances through the co-evolution of policy and reward capabilities, and further incorporates reflection-driven data generation, which strengthens the integration between reasoning and judgment and brings additional performance improvements.

Table 2: **Evaluation Results on SPARK-7B.** We evaluate SPARK on multiple mathematical and reward-related benchmarks. Here, RB2 denotes RewardBench2.

| Model | AIME24 | AIME25 | AMC23 | GSM8k | Math-500 | MMLU-STEM | Avg-M | RB2 | RB2-Math | Avg-R | Avg-All |
|---|---|---|---|---|---|---|---|---|---|---|---|
| | **LLM Math Benchmark** | | | | | | | **Reward Benchmark** | | | |
| *Baseline* | | | | | | | | | | | |
| Qwen2.5-7B | 6.7 | 6.7 | 50.0 | 91.9 | 76.2 | 75.8 | 51.2 | 49.5 | 41.0 | 45.3 | 49.7 |
| Simple-RL-Zero | 16.7 | 6.7 | **62.5** | 92.0 | 78.0 | 64.5 | 53.4 | 31.0 | 38.7 | 34.9 | 48.8 |
| Eurus-2-7B-PRIME | **26.7** | - | 57.8 | - | 79.2 | 71.0 | - | - | - | - | - |
| Open-Reasoner-Zero-7B | 13.3 | - | 47.0 | - | 79.2 | 70.3 | - | 31.4 | 37.3 | 34.4 | - |
| Qwen2.5-Math | 13.3 | - | 50.6 | - | **79.8** | 60.2 | - | - | - | - | - |
| *Qwen2.5-7B + GRPO* | | | | | | | | | | | |
| + Policy-Only | 6.7 | 6.7 | 52.5 | 92.4 | 76.0 | 76.2 | 51.8 | 48.4 | 40.4 | 44.4 | 49.9 |
| + Reward-Only | 13.3 | 3.3 | 50.0 | 90.2 | 73.6 | 79.3 | 51.6 | 43.6 | 40.4 | 42.0 | 49.2 |
| + Policy&Reward | 16.7 | 3.3 | 50.0 | 92.7 | 76.6 | 80.1 | 53.2 | 48.3 | 43.7 | 46.0 | 51.4 |
| *Ours* | | | | | | | | | | | |
| SPARK-7B | 16.7 | **6.7** | **62.5** | **93.2** | 79.4 | **81.1** | **56.6** | **58.8** | **55.7** | **57.3** | **56.8** |
| Δ | **+10.0** | **+0.0** | **+12.5** | **+1.3** | **+3.2** | **+5.3** | **+5.4** | **+9.3** | **+14.7** | **+12.0** | **+7.1** |

Table 3: **Evaluation Results on SPARK-VL-32B.** We evaluate SPARK on multiple mathematical and reward-related benchmarks. Here, RB2 denotes RewardBench2, and VL-RB denotes VL-RewardBench.

| Model | MathVista | MathVision | WeMath | MathVerse | DynaMath | LogicVista | MMK12 | Avg-M | RB2 | RB2-Math | VL-RB | VL-RB-Math | Avg-R | Avg-All |
|---|---|---|---|---|---|---|---|---|---|---|---|---|---|---|
| | **VLM Math Benchmark** | | | | | | | | **Reward Benchmark** | | | | | |
| *Baseline* | | | | | | | | | | | | | | |
| Qwen2.5-VL-32B | 74.7 | 38.4 | 69.1 | 48.5 | 61.3 | 55.4 | 52.9 | 57.2 | 57.0 | 59.6 | 59.2 | 56.0 | 58.0 | 57.5 |
| VL-ReThinker-32B | 78.8 | **40.5** | 76.7 | 56.9 | **62.9** | 51.8 | 72.9 | 62.9 | 53.9 | 51.8 | 49.9 | 23.5 | 44.8 | 56.3 |
| Vision-R1-32B | 73.2 | 35.7 | **78.9** | 53.8 | **62.9** | 54.2 | 55.2 | 59.1 | 53.4 | **68.9** | - | - | - | - |
| MM-Eureka-32B | 74.8 | 34.4 | 73.4 | 56.5 | 62.1 | 53.4 | 72.2 | 61.0 | 56.4 | 58.5 | 58.3 | 56.6 | 57.5 | 59.7 |
| *Qwen2.5-VL-32B + GRPO* | | | | | | | | | | | | | | |
| + Policy&Reward | 78.2 | 40.2 | 77.1 | 57.7 | 60.9 | 57.4 | 72.3 | 63.4 | 56.3 | 56.4 | 57.1 | 53.6 | 55.9 | 60.7 |
| *Ours* | | | | | | | | | | | | | | |
| SPARK-VL-32B | **79.1** | 40.2 | 76.7 | **59.2** | **62.9** | **59.4** | **77.4** | **65.0** | **60.3** | 62.7 | **61.4** | **59.6** | **61.0** | **63.5** |
| Δ | **+4.4** | **+1.8** | **+7.6** | **+10.7** | **+1.6** | **+4.0** | **+24.5** | **+7.8** | **+3.3** | **+3.1** | **+2.2** | **+3.6** | **+3.0** | **+6.0** |

Notably, from Tab. 1 we also observe that SPARK-VL-7B achieves significant improvements on both reward-related benchmarks, including RewardBench2 (**+3.5%**) and VL-RewardBench (**+15.0%**). Although all reward-related data generated during training are mathematics-specific, these two reward benchmarks span diverse domains. This indicates that SPARK is able to generalize its judgment ability beyond mathematics and perform strongly on broader tasks.

Overall, these results demonstrate that policy and reward are not in competition but instead mutually reinforcing. Joint optimization, when augmented with reflection, produces a synergistic effect that drives simultaneous and significant improvements in both reasoning and judgment. Representative reasoning cases on mathematical are illustrated in Fig. 3. More cases can be found in Appendix A.4.

**Results on SPARK-7B** To further assess the generalizability of our approach, we conduct additional experiments on the LLM Qwen2.5-7B. As shown in Tab. 2, SPARK-7B achieves average improvements of **5.4%** on mathematical benchmarks, **12.0%** on reward benchmarks, and **7.1%** overall, demonstrating consistent gains across diverse evaluation settings.

Models such as Simple-RL-Zero (Zeng et al., 2025) and the *Policy-Only* variant of Qwen2.5-7B show declines on reward benchmarks, which reflects the trade-off between reasoning and judgment ability. Moreover, we observe that *Reward-Only* GRPO training performs worse than *Policy-Only* training on both reasoning and reward tasks. This suggests that the model tends to overfit to reward signals, which in turn weakens its reasoning ability and prevents it from excelling in either skill. The *Policy&reward* GRPO variant partially alleviates this issue by training both capabilities simultaneously. Building on this, SPARK-7B further leverages on-policy co-evolution and reflection mechanisms to more effectively fuse and reinforce reasoning and judgment, ultimately achieving a substantial performance leap.

Table 4: **Evaluation Results on General Multimodal Benchmarks.** We select multiple general multimodal benchmarks to assess the generalization and robustness of our method.

| Models | MMBench | MMStar | MMMU | MMVet | ScienceQA | POPE | SeedBench | RealWorldQA | Average |
|---|---|---|---|---|---|---|---|---|---|
| Qwen2.5-VL-7B | 82.2 | 64.1 | 58.0 | 69.7 | 89.0 | 85.9 | 77.0 | 68.4 | 74.3 |
| VL-Rethinker-7B | 82.3 | 65.4 | **59.0** | 69.3 | 87.8 | 86.1 | 76.3 | 69.3 | 74.4 |
| MM-Eureka-7B | 84.2 | 65.3 | 57.8 | 68.9 | 88.8 | 85.8 | 76.8 | 65.1 | 74.1 |
| SPARK-VL-7B | **84.4** | **67.3** | 58.7 | **71.5** | **90.8** | **88.2** | **77.2** | **68.5** | **75.8** |
| Δ | **+2.2** | **+3.2** | **+0.7** | **+1.8** | **+1.8** | **+2.3** | **+0.2** | **+0.1** | **+1.5** |

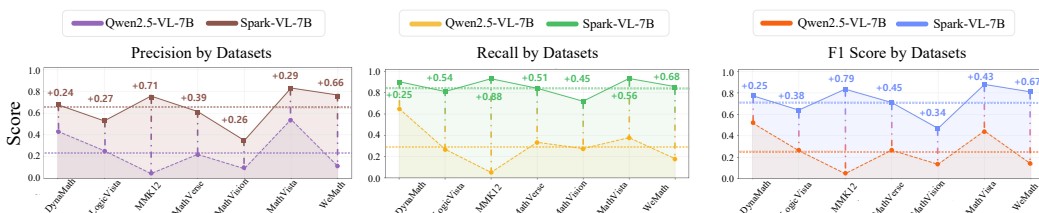

Figure 4: **Study on Model's Reward Accuracy.** We evaluate the model's judgment ability by measuring its accuracy in determining whether its own answers are correct.

**Results on Qwen2.5-VL-32B** To further examine the scalability of our approach, we conduct experiments with Qwen2.5-VL-32B as the backbone. As shown in Tab. 3, SPARK-VL-32B achieves improvements of **+7.8%** on mathematical benchmarks and **+3.0%** on reward benchmarks. These results confirm that SPARK scales effectively to larger models, with the co-evolving mechanism continuing to deliver consistent gains by leveraging the richer capacity of the backbone. This highlights both the robustness of our framework and its potential for application to even stronger models.

## 4.3 RESULTS ON GENERAL BENCHMARKS

We further assess our approach on a suite of general-purpose benchmarks to evaluate its capabilities beyond mathematics and judgment tasks. As presented in Tab. 4, SPARK-VL-7B achieves improvements of **2.2%**, **3.2%**, and **1.8%** on MMBench (Liu et al., 2023), MMStar (Chen et al., 2024), and MMVet (Yu et al., 2023), respectively, leading to an average gain of **1.5%** across eight benchmarks. Compared with other reinforcement learning approaches, SPARK extends its reasoning and reflection abilities beyond the mathematical domain, demonstrating stronger generalization to diverse tasks.

## 4.4 JUDGMENT ACCURACY ANALYSIS

To further compare the judgment ability of SPARK with the Qwen2.5-VL-7B baseline, we conduct an additional evaluation on self-judgment accuracy. Specifically, we use data from seven mathematical datasets as input: the model is required to first perform reasoning and then assess the correctness of its own prediction. Based on these judgments, we compute recall, precision, and F1 scores. The results, shown in Fig. 4, indicate that SPARK consistently outperforms the baseline across all three metrics, with particularly pronounced gains on MMK12 (Meng et al., 2025), MathVerse (Zhang et al., 2024), and WeMath (Qiao et al., 2024). These findings demonstrate that our training framework enables strong self-judgment ability without relying on manually annotated preference data or larger teacher models.

## 4.5 ABLATION STUDIES

**Ablation Study on Test-Time Scaling** In SPARK, we adopt a reflection-augmented test-time scaling (TTS) strategy to activate the model's integrated capabilities of reasoning, judgment, and self-reflection. As shown in Tab. 5, applying TTS to Qwen2.5-VL-7B leads to a noticeable performance drop across multiple benchmarks, primarily because its weak judgment and reflection skills cause frequent misjudgments, especially when the number of reasoning rounds increases. For the *GRPO+Policy* setting, TTS yields only marginal improvements. In contrast, SPARK achieves sub-

Table 5: **Ablation on Test-Time Scaling.** We conduct ablation studies on the TTS. Specifically, we apply judge-reflection-based TTS to both the Qwen2.5-VL-7B model and the GRPO-trained model to evaluate its effectiveness.

| Model | MathVista | MathVision | WeMath | MathVerse | DynaMath | LogicVista | MMK12 | Average |
|---|---|---|---|---|---|---|---|---|
| *Baseline* | | | | | | | | |
| Qwen2.5-VL-7B | 68.2 | 25.1 | 62.1 | 49.2 | 53.3 | 40.4 | 45.1 | 49.1 |
| Qwen2.5-VL-7B+TTS | 68.5 | 18.4 | 24.1 | 29.9 | 52.7 | 43.1 | 42.8 | 39.9 |
| *Qwen2.5-VL-7B + GRPO* | | | | | | | | |
| + Policy | 72.0 | 28.5 | 67.9 | 51.2 | 54.9 | 44.9 | 66.9 | 55.2 |
| + Policy & TTS | 73.1 | 28.7 | 67.5 | 51.9 | 57.7 | 48.9 | 68.9 | 56.6 |
| SPARK-VL-7B | 75.9 | 31.1 | 70.3 | 53.0 | 58.7 | 50.7 | 71.8 | 58.8 |

Table 6: **Ablation Study on Answer- vs. CoT-based Data Generation.** We generate data on-policy using either final answers, chains of thought (CoT), or a combination of both, and evaluate the impact of these strategies on performance.

| Model | VLM Math Benchmark | | | | | | | Avg-M | Reward Benchmark | | | | Avg-R | Avg-All |
|---|---|---|---|---|---|---|---|---|---|---|---|---|---|---|
| | MathVista | MathVision | WeMath | MathVerse | DynaMath | LogicVista | MMK12 | | RB2 | RB2-Math | VL-RB | VL-RB-Math | | |
| SPARK + Ans | 73.8 | 29.1 | 69.0 | 51.9 | 58.1 | 46.2 | 69.9 | 56.9 | 47.2 | 41.8 | 57.9 | 63.9 | 52.7 | 55.3 |
| SPARK + CoT | 73.6 | 29.7 | 67.4 | 50.4 | 57.3 | 48.4 | 71.3 | 56.9 | 52.5 | 44.0 | 62.3 | 60.8 | 54.9 | 56.2 |
| SPARK + Ans&CoT | 75.9 | 31.1 | 70.3 | 53.0 | 58.7 | 50.7 | 71.8 | 58.8 | 49.3 | 39.2 | 62.7 | 65.1 | 54.1 | 57.1 |

stantially larger gains, benefiting from its inherently enhanced reasoning, judgment, and reflection capabilities.

**Ablation Study on Answer- vs. CoT-based Data Generation** In SPARK training, on-policy reward data can be generated either from final answers or from chains of thought (CoT). As shown in Tab. 6, we compare three settings: using only answer-based data (55.3), using only CoT-based data (56.2), and combining both (57.1). The results indicate that integrating both sources leads to the best performance, suggesting that the complementary nature of answer- and CoT-based data provides richer training signals and ultimately enhances the model's learning effectiveness.

**Cost Analysis** Unlike traditional RM-based RL methods, SPARK removes the need for an additional reward model and extra preference data. As shown in Tab. 7, RM-based RL requires a separate RM training stage with large-scale human or teacher annotations, and during RL optimization, it repeatedly calls the RM for reward inference, which doubles GPU usage and slows training. In contrast, SPARK directly employs lightweight rule-based verifi-

Table 7: **Comparison between RM-based RL and SPARK.**

| | RM-based RL | SPARK (Ours) |
|---|---|---|
| **Extra Data (Preference)** | ✓ | ✗ |
| **Extra RM Training** | ✓ | ✗ |
| **GPU Cost** | ~2× | 1× |
| **Reward Signal** | RM inference | Rule-based signal |
| **Efficiency** | Slower | Faster |

able rewards to generate feedback on-policy, allowing a single unified model to optimize both policy and reward. This design not only reduces data and computational costs but also ensures a faster and more scalable training pipeline.

## 5 CONCLUSION

We presented the **S**ynergistic **P**olicy **A**nd **R**eward Co-Evolving Framewor**K** (**SPARK**), an efficient, on-policy, and stable paradigm that unifies policy optimization and reward modeling within a single model. Unlike prior RL pipelines that treat policy and reward in isolation or rely on costly external reward models, SPARK recycles RLVR rollouts into judgment and reflection objectives, enabling the model itself to function as both a strong policy and a generative reward model. This co-evolving mechanism establishes a positive feedback loop: improved reward accuracy enhances reasoning ability, while stronger reasoning in turn refines reward judgment, fostering self-reflection and stability. Demonstrating substantial improvements on mathematical, reward, and general benchmarks, SPARK offers a scalable and generalizable solution for RL, advancing a new paradigm where reasoning, judgment, and reflection evolve synergistically.

ETHICS STATEMENT

This work adheres to the ICLR Code of Ethics. Our study focuses on improving reinforcement learning methods for large language and vision–language models without involving human subjects or sensitive personal data. All datasets used in our experiments are publicly available and widely adopted in the community, and we carefully follow their respective licenses and intended usage. The proposed framework is designed to enhance reasoning, judgment, and reflection capabilities in AI models, and does not introduce new risks of bias, privacy violations, or harmful applications. We are committed to responsible research practices and transparent reporting.

REPRODUCIBILITY STATEMENT

We have made significant efforts to ensure the reproducibility of our work. The paper provides detailed descriptions of our framework, along with the formal definitions and equations. Experimental settings, datasets, and models are specified in the main text and supplementary materials. To further support reproducibility, we will release our training and evaluation code, along with processed datasets and prompts, upon acceptance. These resources allow researchers to replicate all reported results and extend our framework to new tasks.

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

# A    Appendix

## Usage of Large Language Models

In this work, we made limited use of a large language model (LLM) as a writing assistant. Specifically, the LLM was employed solely for language polishing and minor grammatical improvements of the manuscript text. It was not used for research ideation, experimental design, data analysis, or substantive content generation. All technical contributions, results, and conclusions were developed entirely by the authors.

## Outline

In the appendix, we provide additional supporting materials to facilitate a deeper understanding of our work. First, in Sec. A.1, we summarize all the models, datasets, and benchmarks used in the experiments of SPARK. Second, in Sec. A.2, we present the prompt templates employed in our study, including those used during evaluation as well as the on-policy prompt designs adopted by SPARK for reward and reflection data generation. Third, Sec. A.3 discusses related works on self-reward and self-reflection. Finally, in Sec. A.4, we provide several illustrative reasoning cases from SPARK on mathematical and reward benchmarks.

### A.1    Model, Dataset and Benchmark Statistic

#### A.1.1    Models

In our study, we adopt the Qwen family of models as the backbone, including Qwen2.5-VL-7B (Bai et al., 2025), Qwen2.5-VL-32B (Bai et al., 2025), and Qwen2.5-7B (Yang et al., 2025a). Based on these backbones, we train three corresponding variants of our proposed framework: SPARK-VL-7B, SPARK-VL-32B, and SPARK-7B. These variants allow us to comprehensively evaluate the effectiveness of SPARK across both multimodal and text-only settings, as well as across different model scales.

For comparison, we benchmark against a wide range of existing RL-based approaches. In the multimodal domain, we include VL-Rethinker-7B (Wang et al., 2025), MM-Eureka-7B (Meng et al., 2025), OpenVLThinker-7B (Deng et al., 2025), Vision-R1-7B (Huang et al., 2025), and R1-OneVision-7B (Yang et al., 2025b), all of which represent recent efforts to strengthen reasoning capacity in vision–language models through reinforcement learning. In the language domain, we compare with Qwen2.5-Math-7B-Instruct (Yang et al., 2024), Simple-RL-Zero-7B (Zeng et al., 2025), Eurus-2-7B-PRIME (Cui et al., 2025), and Open-Reasoner-Zero-7B (Hu et al., 2025), which focus primarily on mathematical or general reasoning tasks within the NLP setting.

To further examine the generalization ability of SPARK across different scales, we additionally evaluate against larger multimodal baselines, including VL-Rethinker-32B (Wang et al., 2025), MM-Eureka-32B (Meng et al., 2025), and Vision-R1-32B (Huang et al., 2025). These larger-scale models provide an important reference point to test whether the improvements introduced by SPARK are preserved when scaling up.

A complete summary of all models used in our experiments, along with their categories (multimodal vs. language-only, 7B vs. 32B scale), is provided in Tab. 8 of the supplementary material.

#### A.1.2    Benchmarks

We evaluate SPARK across three major categories of benchmarks: mathematical reasoning, reward-related evaluation, and general multimodal understanding. This comprehensive setup ensures that our analysis covers not only specialized domains but also broader tasks.

Mathematical Benchmarks. For multimodal mathematical reasoning, we adopt MathVista (Lu et al., 2023), MathVision (Wang et al., 2024), WeMath (Qiao et al., 2024), MathVerse (Zhang et al., 2024), DynaMath (Zou et al., 2024), LogicVista (Xiao et al., 2024), and MMK12 (Meng et al., 2025). For text-only reasoning, we include AIME24, AIME25, AMC23, GSM8k (Cobbe et al., 2021), Math500 (Lightman et al., 2023), and the MMLU_STEM (Hendrycks et al., 2020). These datasets

Table 8: **Model Sources.** We have compiled a list of all the models involved in the experiments along with their parameter scale.

| Models | Parameter |
|---|---|
| *Baseline* | |
| Qwen2.5-VL-7B-Instruct (Bai et al., 2025) | 7B |
| Qwen2.5-VL-32B-Instruct (Bai et al., 2025) | 32B |
| Qwen2.5-7B-Instruct (Yang et al., 2025a) | 7B |
| *Multimodal* | |
| VL-Rethinker-7B (Wang et al., 2025) | 7B |
| VL-Rethinker-32B (Wang et al., 2025) | 32B |
| MM-Eureka-7B (Meng et al., 2025) | 7B |
| MM-Eureka-32B (Meng et al., 2025) | 32B |
| OpenVLThinker-7B (Deng et al., 2025) | 7B |
| Vision-R1-7B (Huang et al., 2025) | 7B |
| Vision-R1-32B (Huang et al., 2025) | 32B |
| R1-OneVision-7B (Yang et al., 2025b) | 7B |
| *Language-Only* | |
| Qwen2.5-Math-7B-Instruct (Yang et al., 2024) | 7B |
| Simple-RL-Zero-7B (Zeng et al., 2025) | 7B |
| Eurus-2-7B-PRIME (Cui et al., 2025) | 7B |
| Open-Reasoner-Zero-7B (Hu et al., 2025) | 7B |

Table 9: **Benchmark Sources.** We have included information for all the benchmarks tested in the paper in the table.

| Setting | Models |
|---|---|
| **Mathematical Multimodal Benchmark** | MathVista (Lu et al., 2023) |
| | MathVision (Wang et al., 2024) |
| | WeMath (Qiao et al., 2024) |
| | MathVerse (Zhang et al., 2024) |
| | DynaMath (Zou et al., 2024) |
| | LogicVista (Xiao et al., 2024) |
| | MMK12 (Meng et al., 2025) |
| **Reward Benchmark** | RewardBench2 (Malik et al., 2025) |
| | VL-RewardBench (Li et al., 2025) |
| **General Multimodal Benchmark** | MMMU (Yue et al., 2024) |
| | MMVet (Yu et al., 2023) |
| | MMBench (Liu et al., 2023) |
| | MMStar (Chen et al., 2024) |
| | POPE (Li et al., 2023) |
| | ScienceQA (Lu et al., 2022) |
| | SeedBench (Ying et al., 2025) |
| | RealWorldQA |

collectively test numerical reasoning, symbolic manipulation, logical deduction, and competition-style problem solving under both textual and multimodal settings.

Reward-related Benchmarks. We adopt RewardBench2 (RB2) (Malik et al., 2025) and VL-RewardBench (VL-RB) (Li et al., 2025) as representative reward evaluation benchmarks. Both focus on assessing models' ability to judge correctness and quality of generated outputs. In addition, we separately report results on the mathematical subsets of these benchmarks, in order to analyze the interplay between reward judgment and mathematical reasoning.

General-purpose Multimodal Benchmarks. To assess generalization and robustness beyond math and reward domains, we evaluate on a diverse set of multimodal benchmarks, including MMMU (Yue et al., 2024), MMVet (Yu et al., 2023), MMBench (Liu et al., 2023), MMStar (Chen et al., 2024), POPE (Li et al., 2023), ScienceQA (Lu et al., 2022), SeedBench (Ying et al., 2025),

---

**Prompt for Mathematical Evaluation**

"Hint: Please answer the question requiring an integer answer and provide the final value, e.g., 1, 2, 3, at the end.\nQuestion:
How many objects are preferred by more than 90 percent of people in at least one category?
**Please first conduct reasoning, and then answer the question. Repeat the final answer using a '\\boxed{}'.**"

---

Figure 5: **Mathematical Prompt.** Prompt suffix used for mathematical benchmark evaluation.

---

**Prompt for VL-RewardBench Evaluation**

You are a highly capable multimodal AI assistant tasked with evaluating answers to visual questions. Please analyze the
following image and question, then determine which of the two provided answers is better.

Question: {query}

Answer 1: {answers[0]}

Answer 2: {answers[1]}

Please evaluate both answers based on the following criteria:
1. Accuracy: How well does the answer align with the visual information in the image?
2. Completeness: Does the answer fully address all aspects of the question?
3. Clarity: Is the answer easy to understand and well-articulated?
4. Relevance: Does the answer directly relate to the question and the image?

After your evaluation, please:
1. Explain your reasoning for each criterion.
2. Provide an overall judgment on which answer is better (Answer 1 or Answer 2). For example: Overall Judgment: Answer X is
better.

Your response should be structured and detailed, demonstrating your understanding of both the visual and textual elements
of the task.

---

Figure 6: **Prompt for VL-RewardBench.** In the figure, the placeholders *query* and *answer* should be replaced
with the specific content of each task.

and RealWorldQA. These benchmarks cover a wide spectrum of multimodal reasoning and under-
standing tasks, ranging from knowledge-intensive QA to real-world perception challenges.

A complete statistical summary of all benchmarks is provided in Tab. 9.

### A.1.3 TRAINING DATA PREPARATION

Previous approaches that aimed to train a model's judgment or reflection capabilities typically relied
on either reward data generated by a teacher model or manually annotated reflection traces. In
contrast, SPARK is able to generate the required reward and reflection training data on-policy during
policy's RFT, guided by verifiable reward signals. This approach not only greatly reduces the cost
of data collection, but also ensures that the generated data evolves alongside model optimization and
remains consistently aligned with the model's current policy distribution.

Leveraging SPARK 's ability to generate reward and reflection data on-policy, we only need to
collect VQA triples consisting of images, questions, and answers. In our experiments, 19k randomly
sampled instances from ViRL-39k (Wang et al., 2025) are used to train SPARK-VL-7B, while 24k
difficulty-filtered instances from the same dataset are used for SPARK-VL-32B. For the language-
only variant SPARK-7B, we employ the Simple-RL-Zero-25k dataset (Zeng et al., 2025). All data
are represented in the form of $(q, a, I)$, where $q$ denotes the question, $a$ the ground-truth answer,
and $I$ the corresponding image. Notably, this setup requires no manually annotated reward data,
reflection traces, or judgment-oriented CoT trajectories.

918
919

> **Prompt for On-Policy Data Generation**
>
> You're given a question and two possible answers. (A) and (B) may differ in correctness. Please analyze both answers carefully and decide which one is correct. After reasoning, restate your final choice using \\box{}.\n\n<Questions>:\n{question}\n\n(A) {answer_a}\n(B) {answer_b}\n\n<Judgement>:
>
> You are a specialist in answer evaluation. For the given question and answer, perform logical reasoning to assess the answer's correctness. Then state your judgment as 'Yes' or 'No'. Finally, repeat your judgment inside \\box{}.\n\n<Questions>:\n{question}\n\n<Answers>:\n{answer}\n\n<Judgements>:
>
> You are given a question and a reasoning process (CoT). Your task is to judge whether both the reasoning and the final answer are correct. If correct, summarize the logic and repeat the answer. If not, provide a correct reasoning process and answer.\n\nQuestion:\n{question}\n\nProvided CoT:\n{cot}\n\nPlease first conduct reasoning, and then answer the question. Repeat the final answer using a '\\boxed{}'.
>
> You are given a question and a proposed answer. Judge whether the answer is correct. If it is, briefly confirm it. If it is incorrect, provide correct reasoning and the correct answer.\n\nQuestion:\n{question}\n\nAnswer:\n{answer}\n\nPlease first conduct reasoning, and then answer the question. Repeat the final answer using a '\\boxed{}'.

Figure 7: **On-Policy Data Generation Prompts.** The figure illustrates four different prompt templates used for generating reward and reflection data.

## A.2 PROMPTS

**Evaluation Prompts**   During dataset evaluation, we appended an additional prompt at the end of each mathematical problem to facilitate the separation of reasoning steps from the final answer. To avoid over-constraining the model with rigid output formats (e.g., `<think><answer>`), we instead instructed the model to enclose the final answer within $\box{}$ after completing its reasoning process. A concrete example is provided in Fig. 5.

For reward-related benchmarks, we followed the official evaluation prompts provided by each benchmark. For instance, in VL-RewardBench (Li et al., 2025), we adopted the original prompt format as illustrated in Fig. 6.

**On-Policy Data Generation Prompts**   A key step in the policy–reward co-evolving training of SPARK is the on-policy generation of reward and reflection data. During GRPO optimization, the reward signals serve two purposes: on the one hand, they are used to compute the advantage for updating model parameters; on the other hand, they guide the construction of reward and reflection data. Specifically, chain of thought (CoT) or final answers filtered by reward values are wrapped with carefully designed prompts and reorganized into new training samples. These samples further enhance the model's judgment and reflection abilities. Examples of the prompts used for reward and reflection data generation are shown in Fig. 7.

## A.3 RELATED WORKS

**Self-Reward and Self-Reflection.**   Early studies have explored incorporating *self-reward* and *self-reflection* into supervised fine-tuning (SFT) pipelines by generating reward or reflection data to enhance reasoning abilities. For example, STaR (Zelikman et al., 2022) iteratively generates chain-of-thought traces to improve its own reasoning capability. S2R (Ma et al., 2025) employs pre-annotated self-verification and self-correction data for both SFT and RL training. COOPER (Hong et al., 2025) leverages an external assistant to generate preference data, which are then used to train a reward model. While these approaches demonstrate the potential of self-reward and self-reflection for improving reasoning, they still rely on either external annotation data or independently trained reward models. In contrast, our method is the first to unify policy and reward capabilities within a single model by optimizing the GRPO framework. This co-evolving design breaks the conventional paradigm of separately trained reward models, enabling policy and reward to mutually reinforce each other, and integrates reasoning, judgment, and self-reflection into a unified process—without the need for preference annotation or external reward modeling.

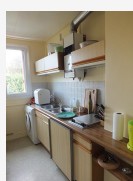

You are a highly capable multimodal AI assistant tasked with evaluating answers to visual questions. Please analyze the following image and question, then determine which of the two provided answers is better......

**Choice:**

["A kitchen counter with a sink, blue plate, and knife block, along with cabinets below and above it; a dishwasher at the base of the cabinetry; a window with curtains on the left side; a towel rack on the wall to the right of the sink; a bottle on the counter; and a silver exhaust hood above the stove.", "A clean and well-organized kitchen showcases a stainless steel gas cooktop, positioned centrally on the counter. Nearby, a stack of plates rests on a shelf above the dishwasher. The wall-mounted air vent is to the left of the sink and gas stove, while the paper towel holder containing white paper towels sits on the right, closer to the dishwasher. A chopping board with a knife lies on the counter, next to a green coffee mug and a bottle, both placed near the edge of the counter. The window's view reveals a glimpse of trees in the distance."]

**Judgement:**

To evaluate the answers, let's break down each criterion:\n\n1. **Accuracy**:\n  - **Answer 1**: This answer accurately describes the kitchen counter with a sink, blue plate, and knife block. It mentions cabinets, a dishwasher, a window, a towel rack, a bottle on the counter, and a silver exhaust hood. However, it does not mention the presence of a stove or a gas cooktop, which is a significant omission.\n  - **Answer 2**: This answer includes a stainless steel gas cooktop, a stack of plates, a wall-mounted air vent, a paper towel holder, a chopping board, a green coffee mug, and a bottle. It also correctly identifies the window and its view. However, it does not mention the sink, the blue plate, or the knife block, which are present in the image.\n\n......4. **Relevance**:\n  - Both answers are relevant to the question and the image, but Answer 1 is more accurate and complete.\n\n**Overall Judgment**: Answer 1 is better.\n\nThe reasoning is that Answer 1 is more accurate and complete, providing a clearer and more detailed description of the kitchen's contents and layout without including any inaccuracies. Answer 2, while attempting to be more descriptive, introduces errors and omissions that do not align with the visual information in the image.

Figure 8: **Reward Reasoning Case.** The example is taken from VL-RewardBench (Li et al., 2025).

## A.4 CASE STUDY

In Fig. 9, we present several reasoning cases on mathematical problems, which provide an intuitive demonstration of SPARK 's integrated capabilities in reasoning, judgment, and reflection. The selected examples highlight scenarios where the model engages in both reasoning and self-judgment.

Furthermore, in Fig. 8, Fig. 10 and Fig. 11, we showcase cases from VL-RewardBench. These examples demonstrate that the judgment ability acquired by our method in mathematical tasks can directly transfer and generalize to broader visual domains.

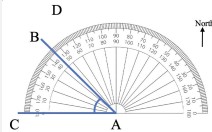

**Prompt:**
The letter T is formed by placing two $2\times 4$ inch rectangles next to each other, as shown. What is the perimeter of the T, in inches?\n\n
Please first conduct reasoning, and then answer the question. Repeat the final answer using a '\\boxed{}'.

<Reasoning>
To determine the perimeter of the letter T formed by placing two \\(2 \\times 4\\) inch rectangles next to each other, we need to visualize the shape and calculate the total perimeter step-by-step.\n\n1. **Visualize the T Shape:**\n  - When two \\(2 \\times 4\\) inch rectangles are placed next to each other, they form a T shape.\n  - The T shape has a vertical rectangle on the right and a horizontal rectangle on top.\n  - The vertical rectangle is \\(4\\) inches wide and \\(2\\) inches tall.\n  - The horizontal rectangle is \\(2\\) inches wide and \\(4\\) inches tall.\n\n2. **Calculate the Perimeter:**\n  - The perimeter of a shape is the sum of the lengths of all its sides.\n  - For the T shape, the perimeter is the sum of the lengths of the outer edges.\n  - The T shape has two vertical sides of \\(4\\) inches each, two horizontal sides of \\(2\\) inches each, and one additional side that is the overlap of the two rectangles, which is \\(2\\) inches.\n  - Therefore, the perimeter is calculated as follows:\n   - Two vertical sides: \\(4 + 4 = 8\\) inches\n   - Two horizontal sides: \\(2 + 2 = 4\\) inches\n   - One additional side (overlap): \\(2\\) inches\n  - Adding these together gives: \\(8 + 4 + 2 = 14\\) inches\n\nThe perimeter of the T is \\(\\boxed{16}\\) inches.
<Judgement>
The given answer is 16 inches. The answer is incorrect.\n\nTherefore, the judgment is \\boxed{No}."
<Reflection>
To determine the perimeter of the letter T formed by placing two \\(2 \\times 4\\) inch rectangles next to each other, we need to calculate the total perimeter of the resulting shape.\n\n1. **Identify the dimensions of the T-shaped figure:**\n  - The T-shaped figure consists of two \\(2 \\times 4\\) inch rectangles placed one above the other.\n  - The top rectangle has a length of 4 inches and a width of 2 inches.\n  - The bottom rectangle also has a length of 4 inches and a width of 2 inches.\n\n2. **Calculate the perimeter:**\n  - When the two rectangles are placed next to each other, the top rectangle's right side and the bottom rectangle's left side will be aligned, forming a straight line.\n  - The perimeter of the T-shaped figure will be the sum of the outer edges of the two rectangles, minus the overlapping edges.\n  - The total perimeter can be calculated as follows:\n   - The top rectangle contributes 2 lengths (4 inches each) and 2 widths (2 inches each) to the perimeter.\n   - The bottom rectangle contributes 2 lengths (4 inches each) and 2 widths (2 inches each) to the perimeter.\n   - However, the overlapping edges (the 4-inch sides of the rectangles) are not part of the perimeter of the T-shaped figure.\n  - Therefore, the perimeter is \\(2 \\times 4 + 2 \\times 2 + 2 \\times 4 + 2 \\times 2 - 4 = 8 + 4 + 8 + 4 - 4 = 20\\) inches.\n\nThe correct final answer is \\(\\boxed{20}\\).

**Prompt:**
As shown in the diagram, there are several points around the protractor. Which point is located to the northwest of point A?\nA. Point B; B. Point C; C. Point D; D. No correct answer.
Please first conduct reasoning, and then answer the question. Repeat the final answer using a '\\boxed{}'.

<Reasoning>
To determine which point is located to the northwest of point A, we need to understand the direction indicated by the protractor. The protractor is oriented with the north direction at the top, which means that angles measured clockwise from the north direction will give us the directions.\n\n1. Identify the direction of point B relative to point A:\n  - Point B is located at an angle of 30 degrees clockwise from the north direction.\n  - Therefore, point B is located 30 degrees east of north.\n\n2. Identify the direction of point C relative to point A:\n  - Point C is located at an angle of 0 degrees clockwise from the north direction.\n  - Therefore, point C is directly north of point A.\n\n3. Identify the direction of point D relative to point A:\n  - Point D is located at an angle of 45 degrees clockwise from the north direction.\n  - Therefore, point D is located 45 degrees north of east.\n\nSince the northwest direction is 45 degrees north of west, and point D is 45 degrees north of east, point D is not in the northwest direction. Point B is 30 degrees east of north, which is not northwest. Point C is directly north, which is not northwest. Therefore, none of the points are located to the northwest of point A.\n\nThe final answer is **\\boxed{D}.**",
<Judgement>
The given answer D is incorrect.\n\n**\\boxed{No}"
<Reflection>
To determine which point is located to the northwest of point A, we need to understand the direction indicated by the protractor and the compass rose.\n\n1. The protractor is oriented with the top of the protractor aligned with the north direction. This means that the angles measured from the north direction will give us the directions of the points relative to north.\n2. Point A is the origin of the protractor, and we need to find a point that is northwest of A. Northwest is a direction that is 45 degrees west of north.\n3. Looking at the protractor, we see that point B is located at an angle of 45 degrees from the north direction. This means that point B is indeed northwest of point A.\n4. Point C is located at an angle of 90 degrees from the north direction.\n5. Point D is located at an angle of 30 degrees from the north direction.\n\nSince point B is the only point that is northwest of point A, the correct answer is:\n\n**\\boxed{A}"

Figure 9: **Math Inference Case.**

**Prompt:**

You are a highly capable multimodal AI assistant tasked with evaluating answers to visual questions. Please analyze the following image and question, then determine which of the two provided answers is better.

**Question:**

How do the elements in the image relate to each other in terms of positioning or composition?

Answer 1:

"In the image, a child is reaching out and attempting to catch a frisbee. The child's positioning suggests that they are standing on a sidewalk, while their arm extends out in an attempt to grasp the flying disc....In addition to the child and frisbee, there is a person near the end of the sidewalk, possibly observing or involved in the play. There is also a car parked further away from the main action, providing context of the urban environment surrounding the scene."

Answer 2:

"In the image, a young boy wearing blue jeans and a black jacket is actively engaged in catching a red frisbee. He is positioned on a sidewalk that stretches across the scene, with another person walking further down the pathway. .... A fire hydrant can also be seen near one of the parked cars, indicating the presence of urban infrastructure. Additionally, there's a trash bag placed behind the first car on the right side of the image. Overall, these elements come together to create an active and lively street scene centered around the boy playing frisbee."

Please evaluate both answers based on the following criteria:
1. Accuracy: How well does the answer align with the visual information in the image?
2. Completeness: Does the answer fully address all aspects of the question?
3. Clarity: Is the answer easy to understand and well-articulated?
4. Relevance: Does the answer directly relate to the question and the image?

After your evaluation, please:
1. Explain your reasoning for each criterion.
2. Provide an overall judgment on which answer is better (Answer 1 or Answer 2). For example: Overall Judgment: Answer X is better.

Your response should be structured and detailed, demonstrating your understanding of both the visual and textual elements of the task.

**Spark Judgement:**

### Evaluation Criteria\n\n

#### **1. Accuracy**\n- **Answer 1**: \n  - The description of the child attempting to catch a frisbee is accurate.\n  - The mention of a person near the end of the sidewalk is somewhat speculative, as the image does not clearly show another person walking.\n  - The car parked further away is accurately described, providing context for the urban environment.\n- **Answer 2**:\n  - The description of the child (young boy in blue jeans and a black jacket) is accurate.\n  - The frisbee is correctly identified as red and positioned above the child's head.\n  - The sidewalk, another person walking further down the pathway, and the parked cars are accurately described.\n  - The mention of a fire hydrant and a trash bag introduces elements not clearly visible in the image, which reduces accuracy.\n\n**Reasoning**: Answer 1 is slightly more accurate because it avoids introducing elements (like the fire hydrant and trash bag) that are not clearly visible in the image. However, both answers are generally accurate in describing the main elements.\n\n

#### **2. Completeness**\n- **Answer 1**:\n  - Focuses on the child, the frisbee, and the urban environment (sidewalk, car).\n  - Mentions a person near the end of the sidewalk, which adds context but is not explicitly shown in the image.\n  - Does not mention details like the child's clothing or the color of the frisbee, which are visible in the image.\n- **Answer 2**:\n  - Provides detailed descriptions of the child's clothing, the frisbee's color, and the surrounding environment (sidewalk, cars, fire hydrant, trash bag).\n  - Includes more specific details about the scene, such as the parallel street and the positioning of objects.\n  - While some elements (fire hydrant, trash bag) are not clearly visible, the answer is more comprehensive in its description.\n\n**Reasoning**: Answer 2 is more complete as it provides detailed descriptions of the child, the frisbee, and the surrounding environment. However, the inclusion of elements not clearly visible in the image slightly detracts from its completeness.\n\n

#### **3. Clarity**\n- **Answer 1**:\n  - The language is clear and easy to understand.\n  - The description is well-structured, focusing on the child's action and the urban setting.\n  - The flow of the answer is smooth, making it easy to follow.\n- **Answer 2**:\n  - The language is also clear and easy to understand.\n  - The description is detailed and provides a vivid picture of the scene.\n  - The answer is well-structured, but the inclusion of elements not clearly visible in the image (fire hydrant, trash bag) slightly complicates the clarity.\n\n**Reasoning**: Both answers are clear and well-articulated. However, Answer 1 is slightly clearer because it avoids introducing elements that are not clearly visible in the image.\n\n

#### **4. Relevance**\n- **Answer 1**:\n  - Directly addresses the question by focusing on the relationship between the child, the frisbee, and the urban environment.\n  - The description of the child's action and the surrounding elements is relevant to the question.\n  - The mention of a person near the end of the sidewalk is somewhat tangential but still related to the scene.\n- **Answer 2**:\n  - Addresses the question by describing the child's action, the frisbee, and the surrounding environment.\n  - Introduces elements (fire hydrant, trash bag) that are not clearly visible in the image, which reduces relevance to the question.\n  - The detailed description of the scene is relevant but includes unnecessary elements.\n\n**Reasoning**: Answer 1 is more relevant because it focuses on the elements that are clearly visible in the image and directly relates to the question. Answer 2 introduces elements that are not clearly visible, which reduces its relevance.\n\n

### **Overall Judgment**\n- **Answer 1** is better because it is more accurate, relevant, and clear. It avoids introducing elements that are not clearly visible in the image and focuses on the main elements of the scene (child, frisbee, urban environment) in a straightforward manner.\n- **Answer 2**, while more complete in its description, introduces elements that are not clearly visible in the image, which reduces its accuracy and relevance.\n\n**Final Answer: Answer 1 is better.**"

Figure 10: **VL-RewardBench Inference Case 1.**

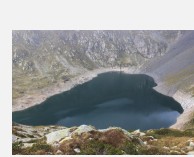

Prompt:
You are a highly capable multimodal AI assistant tasked with evaluating answers to visual questions. Please analyze the following image and question, then determine which of the two provided answers is better.

Question:
Create a detailed description describing the elements of the given worldwide landmark image. Identify the name of the landmark and describe the objects, colors, perspectives, etc.

Answer 1:
"The image showcases a breathtaking view of a mountainous region with a deep, green lake nestled amidst the rocky terrain. The perspective is from a high vantage point, providing a panoramic view of the landscape. The landmark, identified as \"sa_1563\", is a testament to the natural beauty and diversity of our world. The combination of the rugged mountains, the clear blue sky, and the vibrant green lake creates a serene and awe-inspiring scene."

Answer 2:
"The image showcases a serene mountain lake, nestled amidst rocky terrains. The deep green color of the water contrasts with the surrounding landscape, creating a tranquil scene. This landmark is a natural beauty, located in a hilly or mountainous region. The perspective of the image gives a sense of being high above the lake, looking down upon it. However, without specific identifiable features, it's challenging to determine the exact landmark from the description."

Please evaluate both answers based on the following criteria:
1. Accuracy: How well does the answer align with the visual information in the image?
2. Completeness: Does the answer fully address all aspects of the question?
3. Clarity: Is the answer easy to understand and well-articulated?
4. Relevance: Does the answer directly relate to the question and the image?

After your evaluation, please:
1. Explain your reasoning for each criterion.
2. Provide an overall judgment on which answer is better (Answer 1 or Answer 2). For example: Overall Judgment: Answer X is better.

Your response should be structured and detailed, demonstrating your understanding of both the visual and textual elements of the task.

Spark Judgement:
### Evaluation Criteria\n\n

#### **1. Accuracy**\n- **Answer 1**: The description mentions a \"deep, green lake\" and \"rugged mountains,\" which aligns with the image. However, the mention of a \"clear blue sky\" is not evident in the image, as the sky is not visible. The name \"sa_1563\" is provided but is not a recognized landmark name, which introduces an element of inaccuracy.\n- **Answer 2**: This answer accurately describes the deep green color of the water and the rocky terrains surrounding the lake. It also correctly notes the perspective, which appears to be from a higher vantage point. Unlike Answer 1, it avoids mentioning elements (like the sky) that are not present in the image.\n\n**Reasoning**: Answer 2 is more accurate because it sticks to the visual elements present in the image without introducing unnecessary or incorrect details.\n\n

#### **2. Completeness**\n- **Answer 1**: While it provides a detailed description of the landscape, including the lake, mountains, and perspective, it includes the name \"sa_1563,\" which is not a recognized landmark. This adds incomplete or irrelevant information. Additionally, it does not explicitly address the question of identifying the landmark, which is a key part of the prompt.\n- **Answer 2**: This answer focuses on the elements visible in the image, such as the lake, rocks, and perspective. It acknowledges the challenge of identifying the exact landmark due to the lack of specific features, which is a honest and complete response given the information available.\n\n**Reasoning**: Answer 2 is more complete because it addresses the limitations of the image (e.g., the inability to identify the landmark) and focuses on the visible elements without adding unnecessary details.\n\n

#### **3. Clarity**\n- **Answer 1**: The answer is generally clear and well-articulated, with a smooth flow of description. However, the inclusion of the name \"sa_1563\" without explanation can be confusing and detracts from the clarity.\n- **Answer 2**: This answer is also clear and well-structured. It uses simple language to describe the scene and acknowledges the ambiguity regarding the landmark's identity, which adds to its transparency and clarity.\n\n**Reasoning**: Both answers are clear, but Answer 2 is slightly better because it avoids confusion by not introducing unclear terms like \"sa_1563.\"\n\n

#### **4. Relevance**\n- **Answer 1**: The answer is somewhat relevant, as it describes the image's elements. However, the inclusion of the name \"sa_1563\" is irrelevant and distracts from the actual content of the image. The description of the \"clear blue sky\" is also irrelevant since the sky is not visible.\n- **Answer 2**: This answer is highly relevant. It directly addresses the visual elements in the image, such as the lake, rocks, and perspective, and honestly discusses the difficulty of identifying the landmark. It stays focused on the image and the question.\n\n**Reasoning**: Answer 2 is more relevant because it stays focused on the image and the question, avoiding irrelevant details.\n\n

### **Overall Judgment\n**Answer 2 is better.**\n\n**Reasoning**:\n- **Accuracy**: Answer 2 is more accurate as it avoids introducing elements not present in the image (e.g., the sky, the name \"sa_1563\").\n- **Completeness**: Answer 2 is more complete because it honestly addresses the limitations of identifying the landmark and focuses on the visible elements.\n- **Clarity**: Both answers are clear, but Answer 2 is slightly better for its transparency about the unknowns.\n- **Relevance**: Answer 2 is more relevant as it stays focused on the image and the question without introducing irrelevant details.\n\nOverall, Answer 2 provides a more precise, honest, and focused description of the image and its elements."

Figure 11: **VL-RewardBench Inference Case 2.**

