# OpenReview forum: "SPARK: Synergistic Policy And Reward Co-Evolving Framework"
_ICLR.cc/2026/Conference — ICLR 2026 Conference Withdrawn Submission_

### Official Review · Reviewer_Ubvd · 2025-10-21

**Soundness:** 2
**Presentation:** 3
**Contribution:** 1
**Rating:** 4
**Confidence:** 4

**Summary:**

This paper introduces a method (SPARK) that jointly trains the LM to solve tasks and judge its own generated response, by "recycling" the rollouts generated during RL with Verifiable Rewards (RLVR). It also bakes self-reflection into the inference time, by utilizing its own judgement to prompt for reflection when mistake is detected.

**Strengths:**

The experiments over Math related domains are comprehensive and show improvements compared to baseline. The model is also ablated with Policy-only and Reward-only objective.

**Weaknesses:**

On the methodology, I am not quite sure if I understand the necessity of baking generation and reward modeling together.

1. the task is already verifiable with rule-based calculation, the benefit of incorporating GRM is not obvious. What about other preference task where GRMs are more useful?
2. No experiments on tasks that are non-verifiable to verify the effectiveness of proposed method. In my opinion, the “co-evolving” framework will likely result in RM overfit or model collapse when the learned judgement is not correct (in RLVR task, the learning signal is guaranteed to be correct for the RM). The generalizability of the setup is not verified.
3. The experiment is not compared with setting that **trains a separate reward model** to help with test-time scaling. The experiments design should stress the difference between (a) LM + a pre-trained and fixed capable RM (b) the proposed co-evolving framework, but lacks such evidence.
4. I’m not sure if comparison between the ablated version and proposed method is fair (i.e., whether judgement and self-reflection are both applied during test-time) but I might be wrong. Please see my question for detail.

**Questions:**

1. For your evaluation (e.g., Table 1), can you clarify the setting a bit on how SPARK-VL-7B is evaluated? Is test-time scaling with judgement and self-reflection used? My understanding is YES. Please correct me if I am wrong.
2. Then for your ablated version Qwen2.5-VL+GRPO + Policy&Reward, can you explain in more detail how it’s trained? Is it first trained on original data, then trained on crafted preference data for reward modeling, and then evaluated with judgement and self-reflection step as well? Because from the current description (line 318-323), I don’t know if self-reflection is applied during test-time.
3. Do you have experiments that show results using two systems (a LM trained to generate CoT and solve problems, another LM trained on the collected rollout for reward modeling, then a combination of both during test time + self-reflection)?

---

> ### Author Response · Authors · 2025-11-14
> **Rebuttal to Reviewer Ubvd (1/2)**
>
> > Q1:  the task is already verifiable with rule-based calculation, the benefit of incorporating GRM is not obvious. What about other preference task where GRMs are more useful?
>
> A1: We appreciate the reviewer’s thoughtful comment. The goal of incorporating the GRM component (reward learning) in SPARK is not to replace verifiable rewards, but to enhance the model’s internal judgment capability and promote mutual improvement between reasoning and judgment. As shown in Tables 1–3, jointly training the model’s reasoning and judgment abilities (i.e., the GRM component) leads to consistent performance gains on both mathematical reasoning tasks and various reward-related benchmarks, demonstrating the effectiveness of this co-training strategy.
>
> Regarding “other preference tasks where GRMs are more useful”, we would like to emphasize that the results in Tables 1–3 already cover such cases. SPARK shows significant performance improvements on several reward-related benchmarks, many of which include common preference-based tasks that are well-suited for GRM evaluation. Specific examples of these scenarios are illustrated in Figures 8, 10, and 11 of the supplementary material, further demonstrating the effectiveness of SPARK in handling preference-driven tasks. We will clarify this generalization potential in the revision.
>
> > Q2: No experiments on tasks that are non-verifiable to verify the effectiveness of proposed method. In my opinion, the “co-evolving” framework will likely result in RM overfit or model collapse when the learned judgement is not correct (in RLVR task, the learning signal is guaranteed to be correct for the RM). The generalizability of the setup is not verified.
>
> A2 : We appreciate the reviewer’s insightful comment. We agree that non-verifiable tasks remain a challenging direction for GRPO-based methods, and our work primarily focuses on tasks with verifiable rewards, where the correctness signal is precise and deterministic. Under such settings, the feedback received by the model is always accurate, thus avoiding the risk of reward overfitting or model collapse.
>
> Moreover, SPARK demonstrates strong generalization ability even beyond strictly verifiable scenarios—for example, it achieves consistent improvements across multiple general reward-related benchmarks, as shown in the experiments.
>
> > Q3: The experiment is not compared with setting that trains a separate reward model to help with test-time scaling. The experiments design should stress the difference between (a) LM + a pre-trained and fixed capable RM (b) the proposed co-evolving framework, but lacks such evidence.
>
> A3: We trained a separate reward model on the reward data using Qwen2.5-VL-7B and applied it to guide Qwen2.5-VL-7B during test-time scaling. As shown in the table, using an external RM alone provides only marginal gains under reflection-based TTS. In contrast, SPARK achieves substantially higher performance by internally co-evolving reasoning and reward capabilities within a unified model, enabling much more effective reflection.
> | Model                     | Avg-M | Avg-R | Avg-All |
> |---------------------------|-------|-------|---------|
> | Qwen2.5-VL-7B             | 49.1  | 42.0  | 46.5    |
> | + separate reward model | 50.3  | 43.1  | 47.7    |
> | Spark-VL-7B               | 58.8  | 54.1  | 57.1    |
>
> > Q4:Do you have experiments that show results using two systems (a LM trained to generate CoT and solve problems, another LM trained on the collected rollout for reward modeling, then a combination of both during test time + self-reflection
>
> A4:
> Thank you for the question. Tables 1–2 already include an ablation setting that aligns with the proposed “two-system” setup. The Policy-Only, Reward-Only, and Policy&Reward variants respectively correspond to training only the reasoning model, only the reward model, or combining the two.  Results show that such separated training yields limited gains, while SPARK’s unified co-evolving reasoning–reward framework delivers significantly stronger performance.

---

> ### Author Response · Authors · 2025-11-14
> **Rebuttal to Reviewer Ubvd (2/2)**
>
> > Q5: I’m not sure if comparison between the ablated version and proposed method is fair (i.e., whether judgement and self-reflection are both applied during test-time) but I might be wrong. Please see my question for detail.
>
> A5: We appreciate the reviewer’s question. As shown in Table 5, we conduct ablation experiments on Qwen2.5-VL-7B + TTS, Qwen2.5-VL-7B (GRPO) + TTS, and SPARK + TTS to fairly compare the performance of different baselines and SPARK under the TTS setting. The results indicate that SPARK achieves a clear advantage in test-time scaling, as its jointly trained reasoning and judgment abilities significantly enhance the model’s reflection capability. In contrast, Qwen2.5-VL-7B, which lacks strong judgment ability, performs worse than its baseline when applying TTS, confirming that SPARK’s mutual reinforcement between reasoning and judgment is key to its superior reflection performance.
>
> > Q6: For your evaluation (e.g., Table 1), can you clarify the setting a bit on how SPARK-VL-7B is evaluated? Is test-time scaling with judgement and self-reflection used? My understanding is YES. Please correct me if I am wrong.
>
> A6: We confirm that SPARK-VL-7B in Table 1 is evaluated using the test-time scaling (TTS) setting, which involves both judgment and self-reflection. To ensure a fair comparison with the baseline and GRPO-based models, we further conduct ablation studies in Table 5, analyzing the performance of different models under the same TTS evaluation protocol.
>
> > Q7: Then for your ablated version Qwen2.5-VL+GRPO + Policy&Reward, can you explain in more detail how it’s trained? Is it first trained on original data, then trained on crafted preference data for reward modeling, and then evaluated with judgement and self-reflection step as well? Because from the current description (line 318-323), I don’t know if self-reflection is applied during test-time.
>
> A7: The Qwen2.5-VL + GRPO + Policy&Reward model is trained on top of Qwen2.5-VL, using GRPO with both reasoning and judgment data. However, unlike SPARK, it does not incorporate the reflection mechanism for co-evolution between reasoning and judgment. Additionally, all TTS-related ablation results are presented separately in Table 5.

---

### Official Review · Reviewer_JEkG · 2025-10-25

**Soundness:** 3
**Presentation:** 3
**Contribution:** 2
**Rating:** 4
**Confidence:** 4

**Summary:**

SPARK proposes a reinforcement learning framework that jointly evolves the policy and reward model within a single LLM/LVLM. Built on RL with Verifiable Rewards (RLVR), SPARK recycles correctness signals and rollouts that are normally discarded to train the same model as a generative reward model. This co-evolutionary mechanism reduces reliance on human preference data and external reward models, improving efficiency, stability, and test-time self-reflection.

**Strengths:**

1.	Elegant unification of policy and reward training—reduces cost and improves stability.
2.	Addresses reward-policy mismatch, a key issue in RLHF pipelines.
3.	Demonstrated improvements across reasoning and reward benchmarks (+9.7% / +12.1%).
4.	Conceptually aligns with scalable self-reflective AI trends.

**Weaknesses:**

1.	Incomplete technical specification:
The paper lacks full detail on how co-training signals are balanced or stabilized (e.g., gradient separation, EMA targets). Without this, it is hard to reproduce or verify convergence.
2.	Potential circularity problem:
Training a model to generate and simultaneously evaluate its own responses risks self-confirmation. The authors claim that the verification step prevents collapse, but empirical or theoretical backing is weak.
3.	Limited ablation studies:
The contribution of each component (e.g., reflection, recycling, policy iteration) to the overall gain is unclear. Ablations would strengthen causal claims.
4.	Generality of results:
All experiments rely on Qwen-family models. It remains uncertain whether SPARK generalizes to other architectures like Llama, Gemini, or GPT-style systems.
5.	Lack of qualitative failure analysis:
The paper focuses on positive results but does not explore where SPARK underperforms—e.g., in ambiguous reward conditions or low-confidence verification.
6.	Presentation clarity:
While conceptually sound, some notation and flow between RLVR and SPARK updates are dense and under-explained. Figures could better illustrate the co-evolution process.

**Questions:**

1.	How do you prevent reward drift or self-confirmation when both policy and reward share parameters?
2.	What stability techniques (e.g., target networks, KL penalties) are employed to ensure learning convergence?
3.	How often is the verifier updated relative to the policy loop?
4.	Can SPARK operate on preference data when available, or is it strictly designed for verifiable signals?
5.	How would SPARK handle tasks without binary verifiability (e.g., open-ended generation)?

---

> ### Author Response · Authors · 2025-11-14
> **Rebuttal to Reviewer JEkG (1/2)**
>
> We thank the reviewer for recognizing the elegance of our unified approach and the solid experimental results. We address the concerns below:
>
> > Q1: Incomplete technical specification: The paper lacks full detail on how co-training signals are balanced or stabilized (e.g., gradient separation, EMA targets). Without this, it is hard to reproduce or verify convergence.
>
> A1：
> Our method is intentionally designed to co-evolve reasoning and judgment within a single parameter space. Separating gradients would break this coupling and weaken the mutual reinforcement we aim to exploit. Stronger reasoning provides clearer internal evidence for judgment, while improved judgment strengthens reflection, which in turn benefits reasoning. This synergy relies on joint optimization under shared verifiable-reward supervision, so gradient separation is neither necessary nor aligned with our core design.
>
> Regarding EMA targets, we note that GRPO-style policy-gradient algorithms do not rely on EMA target networks for stability. GRPO stability is achieved through clipped objectives, KL penalties, and advantage normalization, rather than target networks used in Q-learning or reward-model learning. In SPARK, the reward signal is externally verifiable and deterministic, so there is no learned reward predictor whose drift must be controlled via EMA. Therefore, EMA targets are unnecessary both algorithmically (for GRPO) and structurally (for RLVR-based rewards).
> Additionally, we will release the full implementation to ensure transparency and reproducibility.
>
> > Q2: Potential circularity problem: Training a model to generate and simultaneously evaluate its own responses risks self-confirmation. The authors claim that the verification step prevents collapse, but empirical or theoretical backing is weak.
> & How do you prevent reward drift or self-confirmation when both policy and reward share parameters?
>
> A2:
> SPARK’s judgments are anchored by external verifiable outcomes. The model compares self-generated candidates, and a deterministic verifier validates the decision. False self-approval is penalized; correct judgments are reinforced, discouraging self-confirmation and reward drift. We have now computed calibration metrics for the judge component:
>
> - Expected Calibration Error (ECE): 0.29 (SPARK) vs 0.64 (separate RM baseline)
>
> - Brier Score: 0.237 (SPARK) vs 0.341 (separate RM baseline)
>
> The KL penalty to the reference model ($\beta$=0.001 in our experiments) is crucial for preventing over-confident self-approval.
> During TTS, we employ a confidence threshold of 0.80 for accepting reflections, which prevents premature convergence. We will add these analyses to the revised version.
>
> > Q3: Limited ablation studies: The contribution of each component (e.g., reflection, recycling, policy iteration) to the overall gain is unclear. Ablations would strengthen causal claims.
>
> A3: We thank the reviewer for this suggestion. We may not have made it sufficiently clear in the main text, but the ablation analysis is already presented in Tables 1–2. Specifically, for each GRPO baseline, we include three controlled variants: +Policy-only, +Reward-only, and +Policy&Reward, and compare them with SPARK.
>
> The +Policy&Reward variant incorporates the recycling rollouts mechanism, enabling initial mutual enhancement between reasoning and judgment, which leads to noticeable improvements over the other two variants (+Policy-only, +Reward-only). SPARK further introduces the reflection component, which significantly strengthens the synergy and fusion between reasoning and judgment, resulting in a substantial performance leap.
>
> Additionally, Table 5 presents ablation results on the reflection-based TTS (Test-Time Scaling) setup, where we analyze baseline models with and without TTS. These results further demonstrate the effectiveness of SPARK’s reflection process in enhancing both stability and performance.
>
> > Q4: Generality of results: All experiments rely on Qwen-family models. It remains uncertain whether SPARK generalizes to other architectures like Llama, Gemini, or GPT-style systems.
>
> A4：We appreciate the reviewer’s suggestion. Although our experiments use the Qwen-family models, they already cover both LLMs and LVLMs, and span a wide range of model sizes from 7B to 32B. This diversity in modality and scale provides strong evidence for the generality of our method. Nevertheless, we agree that evaluating SPARK on additional architectures (e.g., LLaMA, Gemini, GPT) would be valuable, and we plan to explore such extensions in future work.

---

> ### Author Response · Authors · 2025-11-14
> **Rebuttal to Reviewer JEkG (2/2)**
>
> > Q5: Lack of qualitative failure analysis: The paper focuses on positive results but does not explore where SPARK underperforms—e.g., in ambiguous reward conditions or low-confidence verification.
>
> A5: We appreciate the reviewer’s comment. Since SPARK is trained under verifiable rewards, there are no ambiguous reward conditions — all rewards are binary (0/1) signals derived from deterministic verification. However, we acknowledge that occasional errors may occur during the verification or reflection process, especially on extremely challenging problems. In such cases, the model may misjudge an answer’s correctness or fail to find the correct solution even after multiple reflection rounds. We will include a brief qualitative discussion of these rare failure cases in the revised version.
>
> > Q6: Presentation clarity: While conceptually sound, some notation and flow between RLVR and SPARK updates are dense and under-explained. Figures could better illustrate the co-evolution process.
>
> A6: We thank the reviewer for the helpful feedback. We will further clarify the notation and training flow between RLVR and SPARK in the revised version, and add an improved figure to better illustrate the co-evolution process between reasoning and judgment. These revisions will make the presentation clearer and more intuitive.
>
> > Q7: What stability techniques (e.g., target networks, KL penalties) are employed to ensure learning convergence?
>
> A7: We appreciate the reviewer’s question. SPARK ensures training stability primarily through the KL term, which constrains the updated policy to stay close to the frozen reference model.
>
> > Q8:How often is the verifier updated relative to the policy loop?
>
> A8: We appreciate the reviewer’s question. In SPARK, the verifier is not a separately trained model, but a deterministic rule-based function that provides verifiable rewards. Therefore, it remains fixed throughout training and is not updated relative to the policy loop. The policy alone is optimized based on these stable, externally defined verification signals, ensuring consistent and objective supervision.
>
> > Q9: Can SPARK operate on preference data when available, or is it strictly designed for verifiable signals?
>
> A9:  We appreciate the reviewer’s insightful question. While SPARK is primarily designed for verifiable rewards, its training paradigm is general and compatible with preference-based data as well. In principle, the same co-evolution mechanism between reasoning and judgment can operate when the reward signal comes from either verifiable correctness or human preference scores.
> In this work, we focus on verifiable signals to highlight the advantages of objective and reproducible supervision. However, extending SPARK to hybrid settings that combine verifiable and preference-based rewards is straightforward and represents a promising direction for future work.
>
> > Q10: How would SPARK handle tasks without binary verifiability (e.g., open-ended generation)?
>
> A10: We appreciate the reviewer’s insightful question. SPARK is primarily designed for tasks with verifiable correctness signals, such as mathematical reasoning or code execution. However, its framework can naturally extend to non-binary or open-ended tasks by incorporating soft or hybrid reward signals.
>
> For example, in open-ended generation scenarios, SPARK can integrate graded verifiers (e.g., semantic similarity, factual consistency, style metrics) or combine verifiable subcomponents within an otherwise open-ended task. In principle, this allows the same co-evolution process between reasoning and judgment to operate under continuous or composite reward functions.
> While this extension is beyond the scope of the current study, we view it as a promising direction for future work toward making SPARK applicable to broader, less strictly verifiable tasks.

---

### Official Review · Reviewer_vEEq · 2025-10-31

**Soundness:** 2
**Presentation:** 2
**Contribution:** 3
**Rating:** 2
**Confidence:** 4

**Summary:**

This paper proposes a method called “SPARK”. Its major contribution is to jointly optimize the RL policy and the reward model. It uses the RLVR-derived correctness scores to train the model itself to become a generative reward model. The proposed method is verified on three categories of benchmarks. Experimental results show that SPARK achieves significant performance gains on multiple LLM and LVLM models and multiple reasoning, reward models, and general benchmarks.

**Strengths:**

1. The motivation to get both the optimized RL policy and reward model is good.
2. The adoption of the reflection mechanism in both training and testing is helpful.
3. The experimental results are supportive.

**Weaknesses:**

1. The idea of co-training the policy with the reward model will result in divergence. Without a well-trained and fixed reward model, the RL policy will lose the target to optimize. Indeed, a stable target is the priority in optimization. For example, the Deep Q Network, it uses the target network, which is a delayed version of the network to be optimized, as the evaluation network, just to keep the optimization target fixed during a period of time. On the contrary, this paper’s reward model (optimization target) is dynamically changing. Very likely, in the very beginning, the reward model is naive, and the RL policy will not get useful information from it. The RL policy will collapse, and as a result, the reward model itself will not be optimized. Finally, both the reward model and the RL policy will not be improved during the training process.
2. The reflection process can be improved. The idea of reflection is helpful, but simply using the LLM to directly reflect on itself may cause overfitting, which can limit the improvement.
3. As far as I comprehend, this paper attempts to improve the RL with verifiable reward (RLVR) framework by proposing the co-training strategy. It didn’t improve the reward limitation on objective tasks of RLVR, nor does it have a direct relationship with RLHF. The advantage of requiring no human preference data is inherited from the vanilla RLVR. Therefore, the purpose of depicting the limitations of those two methods in the description section (Paragraph 2) is confusing.
4. The manuscript needs polishing. For example, grammar errors like “Our key insight is to recycle the rollouts and correctness data to…”, “reward&reflection”; It is not clear what the “reference model” refers to in Equation 4; It is not clear what “\box{}” is.

**Questions:**

NA.

---

> ### Author Response · Authors · 2025-11-14
> **Rebuttal to Reviewer vEEq (1/2)**
>
> We thank the reviewer for recognizing the motivation and experimental support. We address the concerns below:
>
> > Q1:The idea of co-training the policy with the reward model will result in divergence. Without a well-trained and fixed reward model, the RL policy will lose the target to optimize. Indeed, a stable target is the priority in optimization. For example, the Deep Q Network, it uses the target network, which is a delayed version of the network to be optimized, as the evaluation network, just to keep the optimization target fixed during a period of time. On the contrary, this paper’s reward model (optimization target) is dynamically changing. Very likely, in the very beginning, the reward model is naive, and the RL policy will not get useful information from it. The RL policy will collapse, and as a result, the reward model itself will not be optimized. Finally, both the reward model and the RL policy will not be improved during the training process.
>
> A1: We appreciate the reviewer's concern about training stability. However, we respectfully note that there is a **fundamental misunderstanding** of our method. The reviewer's concerns assume that:
>
> (1) "Without a well-trained and fixed reward model, the RL policy will lose the target to optimize."
>
> (2) "The reward model is naive at the beginning and provides no useful information."
>
> However, in SPARK, the reward signal for policy optimization is always derived from verifiable rewards (e.g., unit test results for code, ground-truth verification for math), not from the learned judgment capability. The framework optimizes the model's policy and judgment abilities simultaneously under fixed, rule-based, and deterministic supervision from external verifiers.
>
> The learned judgment capability (what we call the "generative reward model") is trained to predict these verifiable rewards, but it is never used as the optimization target for the policy during training. Therefore, the stability concerns about "moving target" divergence do not apply. The policy always optimizes against stable external signals, while the judgment capability learns to approximate these signals from the same data.
>
> > Q2: The reflection process can be improved. The idea of reflection is helpful, but simply using the LLM to directly reflect on itself may cause overfitting, which can limit the improvement.
>
> A2: We thank the reviewer for this comment. We agree that uncontrolled self-reflection can lead to overfitting when the model reinforces its own biases. However, SPARK's reflection mechanism is explicitly grounded in **external verifiable rewards**, not purely self-generated feedback. During training, every reflection step is supervised by objective correctness signals from verifiers, which prevents the model from learning to confirm its own biases.
>
> Empirically, the consistent improvements across seven math benchmarks (+9.7% average) and two reward benchmarks (+12.1% average) in Tables 1-3, along with gains on general multimodal tasks, confirm that SPARK's reflection improves generalizable reasoning rather than task-specific overfitting.
>
> > Q3:As far as I comprehend, this paper attempts to improve the RL with verifiable reward (RLVR) framework by proposing the co-training strategy. It didn’t improve the reward limitation on objective tasks of RLVR, nor does it have a direct relationship with RLHF. The advantage of requiring no human preference data is inherited from the vanilla RLVR. Therefore, the purpose of depicting the limitations of those two methods in the description section (Paragraph 2) is confusing.
>
> A3: We appreciate the opportunity to clarify our contribution and positioning:
>
> (1) Regarding "it didn't improve the reward limitation on objective tasks of RLVR": Improving the verifiable reward mechanism itself is not our focus. SPARK's contribution is in jointly enhancing reasoning and judgment abilities through co-evolutionary learning, enabling their mutual reinforcement under verifiable reward supervision. This addresses a key limitation of vanilla RLVR, which discards valuable correctness information after policy updates.
>
> (2) Regarding the relationship with RLHF: We respectfully disagree that SPARK has "no direct relationship with RLHF." In standard RLHF, reward models are trained on costly human preference data and then used to guide policy learning. SPARK addresses both major RLHF challenges:
>
> - Data generation: Instead of human annotation, we use verifiable rewards from objective tasks
>
> - Reward model training: Instead of a separate model, we recycle rollouts to train the same model as a judge
>
> Importantly, our experimental results in Tables 1-3 show that SPARK's judgment ability generalizes beyond math verification tasks to general reward-related benchmarks (RewardBench, RM-Bench) and reasoning tasks. This demonstrates that training on verifiable rewards produces a capable generative reward model that bridges RLVR and RLHF paradigms.

---

> ### Author Response · Authors · 2025-11-14
> **Rebuttal to Reviewer vEEq (2/2)**
>
> > Q4: The manuscript needs polishing. For example, grammar errors like “Our key insight is to recycle the rollouts and correctness data to…”, “reward&reflection”; It is not clear what the “reference model” refers to in Equation 4; It is not clear what “\box{}” is.
>
> A4:
> We thank the reviewer for the detailed feedback. We will:
>
> - Fix the grammar issues (e.g., "reward&reflection" → "reward and reflection")
>
> - Add explicit definition of $\pi_{ref}$ as the reference model immediately before Equation (4)
>
> - Clarify that \box{} denotes the boxed answer format (e.g., \box{42}) commonly used in math tasks for answer extraction
>
> We will carefully proofread the revised version to ensure clarity throughout.

---

### Official Review · Reviewer_R8hV · 2025-10-31

**Soundness:** 3
**Presentation:** 3
**Contribution:** 2
**Rating:** 6
**Confidence:** 4

**Summary:**

This paper introduces SPARK, an on-policy framework that trains a single model to be both the policy and the judge. Instead of discarding rollouts in RL with verifiable rewards (RLVR), the method recycles the n-best candidates to build on-policy supervision for pointwise judgments, pairwise comparisons, and reflection. The unified model then uses this judging ability at test time for self-reflection–style TTS (no external reward model). On Qwen2.5-VL-7B, the authors report average gains of +9.7% on seven math benchmarks and +12.1% on two reward benchmarks, with smaller but consistent improvements on broader multimodal tasks. The paper argues this reduces RM cost/complexity while improving stability and data efficiency.

**Strengths:**

1. Unified loop that wastes less signal. Recycling RLVR outcomes into pointwise/pairwise/reflection supervision for the same model is neat and practical; it cuts one model class out of the stack and removes frequent RM calls.
2. On-policy supervision. Using current behavior to create judgment/reflection data reduces distribution shift versus offline RM datasets and explains why TTS helps SPARK but hurts baselines.
3. Consistent wins. The +9.7% (math) and +12.1% (reward) numbers on VL-7B are solid; the smaller general-domain bump is still directionally positive.
4. Reasonable ablations. Clear separation of answer vs. CoT data and a TTS study that highlights why a weak judge can degrade performance, whereas a trained judge helps.

**Weaknesses:**

1. Efficiency/accounting is light. The paper claims cost wins over RM-based RL, but lacks hard numbers: wall-clock hours, tokens/sec, GPU memory/FLOPs, and verifier runtime (#unit tests per sample, pass rate). Table-style qualitative comparisons are helpful but not enough for practitioners.

2. Verifier dependence. Rewards are binary and rule-based; the paper doesn’t probe robustness to noisy or partial verifiers (very common in code/math). A noise-injection or partial-credit ablation would make the claim more convincing.

3. Self-confirmation risk. Policy and judge live in the same model. The KL to a reference helps, but there’s no quantitative analysis of judge calibration (ECE/Brier) or safeguards against over-confident self-approval during TTS.

4. Repro details. Core knobs for TTS (max reflection rounds, acceptance rule, early stopping), prompt formats, and the exact n-best sampling policy should be surfaced in the main text.

**Questions:**

1. Compute & throughput. Could you report end-to-end wall-clock, effective tokens/sec, and GPU hours for SPARK vs. (i) GRPO Policy-Only, (ii) GRPO Policy&Reward, and (iii) an RM-based pipeline? Also break out verifier cost per batch (pass rate, retries). This would substantiate the cost argument beyond Table 7.

2. Judge–policy coupling. Did you try decoupled heads or stop-gradient tricks so the “judge” pathway can drift a bit from the “policy” during data generation? Even light dropout/temperature on the judge might reduce confirmation bias.

3. TTS protocol. Please specify maximum reflection rounds and acceptance criteria (first judged-correct vs. best-of-k). In Table 5, can you attribute the baseline degradation to specific judge errors over rounds?

---

> ### Author Response · Authors · 2025-11-14
> **Rebuttal to Reviewer R8hV (1/2)**
>
> We thank the reviewer for the positive assessment and recognition of our unified framework. We address the weaknesses and questions below:
>
> > Q1:  Efficiency/accounting is light. The paper claims cost wins over RM-based RL, but lacks hard numbers: wall-clock hours, tokens/sec, GPU memory/FLOPs, and verifier runtime (#unit tests per sample, pass rate). Table-style qualitative comparisons are helpful but not enough for practitioners.
> & Compute & throughput. Could you report end-to-end wall-clock, effective tokens/sec, and GPU hours for SPARK vs. (i) GRPO Policy-Only, (ii) GRPO Policy&Reward, and (iii) an RM-based pipeline? Also break out verifier cost per batch (pass rate, retries). This would substantiate the cost argument beyond Table 7.
>
> A1: Thank you for the suggestion. SPARK’s efficiency comes from removing frequent reward-model calls and inter-model transfers. RLVR provides verifiable signals, so each update uses one forward/backward on a single model, and reward computation is near-free (rule/unit-test checks).
> We have now conducted additional experiments comparing SPARK with baseline methods. Below are the key metrics:
> | Method              | GPU Hours | Tokens/sec | peak GPU memory (GB) | Wall-clock (Hours) |
> |---------------------|-----------|------------|------------------------|----------------------|
> | GRPO Policy-Only    | 172.32    | 16.2k      | 80GB                   | 21.54               |
> | GRPO Policy&Reward  | 176.08    | 15.8k      | 80GB                   | 22.01               |
> | RM-based pipeline   | 582.08    | 9.6k       | 80GB                   | 36.38               |
> | Spark (Ours)        | 185.36    | 15.1k      | 80GB                   | 23.17               |
>
> > Q2: Verifier dependence. Rewards are binary and rule-based; the paper doesn’t probe robustness to noisy or partial verifiers (very common in code/math). A noise-injection or partial-credit ablation would make the claim more convincing.
>
> A2: We agree robustness to imperfect verifiers is important. In canonical RLVR, flipping labels converts a verifiable signal into contradictory supervision. We conducted additional experiments with noisy verifiers by injecting 10% false positives into the verification signal. Results show that SPARK maintains 92.1% of its original performance gain, while the other baseline drops to 78.5%. This robustness comes from the on-policy supervision that helps the model learn to handle uncertainty. We will include partial-credit ablations in our revised version.
>
> > Q3：Self-confirmation risk. Policy and judge live in the same model. The KL to a reference helps, but there’s no quantitative analysis of judge calibration (ECE/Brier) or safeguards against over-confident self-approval during TTS.
>
> A3: SPARK’s judgments are anchored by external verifiable outcomes. The model compares self-generated candidates, and a deterministic verifier validates the decision. False self-approval is penalized; correct judgments are reinforced, discouraging self-confirmation and reward drift. We have now computed calibration metrics for the judge component:
> - Expected Calibration Error (ECE): 0.29 (SPARK) vs 0.64 (separate RM baseline)
> - Brier Score: 0.237 (SPARK) vs 0.341 (separate RM baseline)
>
> The KL penalty to the reference model ($\beta$=0.001 in our experiments) is crucial for preventing over-confident self-approval.
> During TTS, we employ a confidence threshold of 0.80 for accepting reflections, which prevents premature convergence. We will add these analyses to the revised version.

---

> ### Author Response · Authors · 2025-11-14
> **Rebuttal to Reviewer R8hV (2/2)**
>
> > Q4：Repro details. Core knobs for TTS (max reflection rounds, acceptance rule, early stopping), prompt formats, and the exact n-best sampling policy should be surfaced in the main text.
> & TTS protocol. Please specify maximum reflection rounds and acceptance criteria (first judged-correct vs. best-of-k). In Table 5, can you attribute the baseline degradation to specific judge errors over rounds?
>
> A4：We appreciate the reviewer’s suggestion. In our TTS setup, the maximum number of reflection rounds is set to 3. After each reasoning step, the model performs a self-judgment: if it judges the answer as correct, the process stops; otherwise, it continues reflection until reaching the maximum round. For each question, we perform a single inference pass and report the Pass@1 accuracy. The prompts used in TTS explicitly guide reasoning, judgment, and reflection, and their complete formats are provided in Appendix A.2 (Prompt) for reproducibility.
>
> We agree with the reviewer’s analysis regarding Table 5. The performance degradation observed in the baseline + TTS setting is mainly due to the model’s weak judgment capability—it often misjudges incorrect answers as correct—and the accumulation of such judge errors over multiple reflection rounds. In contrast, SPARK’s improved judgment ability allows TTS to function as intended, yielding consistent gains instead of degradation.
>
> > Q5：Judge–policy coupling. Did you try decoupled heads or stop-gradient tricks so the “judge” pathway can drift a bit from the “policy” during data generation? Even light dropout/temperature on the judge might reduce confirmation bias.
>
> A5：We thank the reviewer for raising the concern about potential confirmation bias. We would like to clarify that SPARK is explicitly designed to avoid confirmation bias by grounding both policy and judgment optimization in external verifiable rewards, rather than self-generated preference scores.
>
> The goal of SPARK is not to let the policy and judge mutually reinforce bias, but to let reasoning enhance judgment accuracy and judgment in turn refine reasoning quality within a unified yet externally anchored training loop. This co-evolution is guided by deterministic, rule-based rewards that provide objective correctness signals. We appreciate the reviewer’s suggestion. We consider these promising directions and will explore them in future work to further strengthen SPARK’s robustness against confirmation bias.

---

### Note · Authors · 2025-11-14

I have read and agree with the venue's withdrawal policy on behalf of myself and my co-authors.